# From Pretraining to Pathology: How Noise Leads to Catastrophic Inheritance in Medical Models

**Hao Sun[1], Zhongyi Han[1]\*, Hao Chen[2], Jindong Wang[3], Xin Gao[4], Yilong Yin[1]\***

[1]School of Software, Shandong University    [2]Carnegie Mellon University
[3]Microsoft Research Asia
[4]Computer Science Program, King Abdullah University of Science and Technology

`sunhao_@mail.sdu.edu.cn`, {`zhongyi.han, ylyin`}`@sdu.edu.cn`,
`haoc3@andrew.cmu.edu, jindong.wang@microsoft.com, xin.gao@kaust.edu.sa`

## Abstract

Foundation models pretrained on web-scale data drive contemporary transfer learning in vision, language, and multimodal tasks. Recent work shows that mild label noise in these corpora may lift in-distribution accuracy yet sharply reduce out-of-distribution generalization, an effect known as catastrophic inheritance. Medical data is especially sensitive because annotations are scarce, domain shifts are large, and pretraining sources are noisy. We present the first systematic analysis of catastrophic inheritance in medical models. Controlled label-corruption experiments expose a clear structural collapse: as noise rises, the skewness and kurtosis of feature and logit distributions decline, signaling a flattened representation space and diminished discriminative detail. These higher-order statistics form a compact, interpretable marker of degradation in fine-grained tasks such as histopathology. Guided by this finding, we introduce a fine-tuning objective that restores skewness and kurtosis through two scalar regularizers added to the task loss. The method leaves the backbone unchanged and incurs negligible overhead. Tests on PLIP models trained with Twitter pathology images, as well as other large-scale vision and language backbones, show consistent gains in robustness and cross-domain accuracy under varied noise levels.

## 1  Introduction

The pretrain–fine–tune paradigm (PT-FT) [1] is now central to medical artificial intelligence. Rather than train from scratch on small, domain-specific datasets, practitioners adapt large-scale foundation models [2] that were learned on web-scale images or image–text pairs. PT-FT supports tasks such as disease classification, tumor detection, and report generation, where expert labels are scarce and costly [3, 4]. Vision models like CLIP [5] and its medical variant PLIP [6], developed for general visual understanding, now underpin applications in pathology, radiology, and dermatology [7]. The same pattern holds in biomedical NLP. Models including BioBERT [8], ClinicalBERT [9], and PubMedBERT [10], pretrained on large biomedical corpora, achieve state-of-the-art performance after fine-tuning on clinical named-entity recognition datasets for diseases, drugs, and genes.

Yet the generalization ability of foundation models in real-world medical tasks, especially under distribution shift, remains far from guaranteed. Recent research has explored various fine-tuning strategies to cope with out-of-distribution (OOD) settings, including domain generalization [11, 12], semi-supervised learning [13], label imbalance [14], and annotation noise [15, 16]. However, these approaches often assume that the pretrained representations are structurally sound and transferable, a premise that becomes fragile under severe distribution shift. In high-stakes domains like medicine,

---

* Corresponding authors.

39th Conference on Neural Information Processing Systems (NeurIPS 2025).

pre-training is commonly performed on noisy or weakly aligned corpora (e.g., scraped clinical documents, deidentified radiology reports, or loosely paired image-text datasets), where semantic mismatch and label ambiguity are pervasive. While scaling up data is often considered beneficial for generalization [17], recent findings show that data *quality* and *distributional alignment* are more decisive than volume alone [18]. Spurious correlations and structural noise embedded in pre-trained features can propagate to downstream OOD tasks, leading to unexpected degradation. This phenomenon is referred to as catastrophic inheritance [18, 19]. It raises critical concerns for safety when deploying foundation models in clinical environments subject to distribution shift.

Label noise is a direct window into how pretraining noises migrate to downstream tasks. Recently, seminal studies in natural-image and general-language benchmarks [18] showed a two-sided effect: a small amount of noise (around five percent) can raise in-domain accuracy after transfer, yet even this mild corruption markedly degrades OOD robustness. Whether the same trade-off exists for medical data remains unknown. How to mitigate the catastrophic inheritance in the medical model has never been explored before. Medical tasks are finer-grained and more sensitive to annotation quality than their natural-image counterparts, and large-scale corpora are noisier. For example, PLIP [6] is pretrained on more than two hundred thousand pathology image–text pairs scraped from Twitter, where many captions are informal, incomplete, or mismatched. Chest-X-ray datasets such as CheXpert [4] and MIMIC-CXR [20] rely on automated report parsing that introduces label ambiguity. Foundation models built on these sources, including MedCLIP variants [7] and BioBERT derivatives [8], inherit this noise but are usually evaluated only in-distribution. Understanding how pretraining noise propagates under clinical distribution shift is therefore an open problem.

This paper presents the first study that systematically investigates the latent impact of pre-training label noise in medical foundation models, focusing on how such noise undermines downstream robustness under distribution shift. While the literature on noisy label learning has proposed numerous techniques to train models robustly on corrupted labels [21, 22, 16], these methods primarily address noise in downstream supervised datasets. Our setting is fundamentally different: the label noise originates from the large-scale pre-training stage, where models are trained on noisy image-text pairs or weakly labeled corpora and later deployed in clinical tasks without further modification to their core parameters. Notably, we do not assume any label noise in the downstream data; instead, we investigate how noise is inherited from pre-training data under distribution shifts

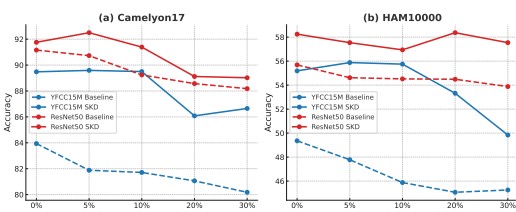

Figure 1: Accuracy of downstream medical classifiers under pre-training label noise. We report performance on Camelyon17 and HAM10000 using features from CLIP and ResNet-50 models pre-trained on ImageNet-1K and YFCC15M with synthetic label corruption. While the presence of label noise can degrade performance, our method (SKD) consistently outperforms the baseline across noise levels and model backbones.

in high-stakes medical applications. The issue is practical, because medical backbones such as PLIP and PubMedBERT are often closed-source, large, and accessed only through frozen checkpoints. Because these models are huge, retraining them is not realistic. We usually freeze the backbone and fine-tune only a lightweight head.

Our study aims to answer the following key questions: 1) *Influence:* Does the noise in pre-training data have an influence on downstream performance in medical settings? 2) *Analysis:* Why does such influence emerge in the representation space? 3) *Mitigation:* How can we mitigate this influence through lightweight fine-tuning? We address these questions on large-scale supervised pre-training, followed by adaptation to downstream medical vision and language tasks.

- **Influence: The label noise in pre-training induces structural degradation in downstream medical tasks.** In Sections 2.1 and 2.2, we conduct controlled experiments with ResNet-50 and CLIP models pre-trained on noisy ImageNet-1K and YFCC15M datasets [23], with label corruption levels ranging from 0% to 30% (0, 5%, 10%, 20%, 30%). We evaluate the resulting models on a suite of medical benchmarks on the downstream OOD tasks. Our findings show that even 5% corruption severely impairs robustness and OOD generalization, including Camelyon17, NIHchestXray, and HAM10000, as shown in Figure 1 and Figure 2.

- **Analysis: The pre-training noise flattens the representational space by reducing skewness and kurtosis.** In Section 2.3, we move beyond spectral analysis and focus on higher-order statistics—specifically, skewness and kurtosis—computed over feature embeddings and logit outputs. As noise levels increase, we observe a consistent decline in both statistics, reflecting a collapse toward less asymmetric and less peaked distributions. This flattening signals reduced expressiveness in the learned representations, which compromises the model's ability to distinguish fine-grained medical categories. These trends are consistently observed across model types and downstream datasets.

- **Mitigation: We propose a distribution-aware fine-tuning strategy that regularizes skewness and kurtosis to counteract representational collapse caused by noisy pre-training.** In Section 7, motivated by the observed flattening of the feature space, we design a light-weight fine-tuning algorithm that encourages asymmetry and peakedness in the downstream representations by explicitly regularizing higher-order statistics. We demonstrate its effectiveness on noisy ResNet-50 and CLIP backbones through extensive evaluations, as shown in Figure 1. In Section 8, we further apply our method to PLIP and other medical foundation models and observe consistent improvements on diverse histopathology benchmarks, highlighting the generality and medical relevance of our approach.

Beyond our core analysis, we argue that this direction is especially important in the medical domain, where pre-trained models are increasingly treated as immutable backbones—either due to their scale, lack of transparency, or access constraints. In such scenarios, fine-tuning often becomes the only controllable lever, and understanding how to correct or compensate for inherited noise is crucial for safe deployment. We believe our findings can inform future work in broader high-stakes settings, including diagnostic support systems, autonomous surgical navigation, and beyond, where noise-induced collapse in representations can have severe real-world implications.

## 2 Understanding the Label Noise in Pre-trained Models

### 2.1 Experiments Design

**Noisy pre-training datasets.** We assume that the supervised pre-training dataset consists of input-label pairs $\mathcal{D} = \{(x_i, y_i)\}_{i \in [N]}$ of size $N$ with accurate supervisions. In practice, $y$ can refer to either a hard label for classification [24, 25] or a text description used in contrastive image-text training [23, 5]. Due to the scale of web-scale corpora and the high cost of expert annotation—particularly in domains like medical imaging—pre-training datasets often contain noisy supervision $\hat{y}$ that does not accurately reflect the true semantics of input $x$ [16, 26]. We define such noisy datasets as $\widehat{\mathcal{D}} = \{(x_i, \hat{y}_i)\}_{i \in [N]}$, and denote $\gamma$ as the ratio of noisy supervision in $\widehat{\mathcal{D}}$.

**Pre-trained models.** We adopt standard backbone architectures that serve as the foundation for downstream tasks, composed of a feature extractor and a projection head. Let $f_\phi : \mathcal{X} \to \mathcal{F}$ denote the feature extractor parameterized by $\phi$, and $g_\theta : \mathcal{F} \to \mathcal{Y}$ denote the projection head. We consider two representative pre-training paradigms: (1) fully supervised classification, where $y$ is the class label and $g_\theta$ is a linear classifier [25]; and (2) contrastive learning, where $g_\theta$ aligns image and text pairs via contrastive objectives [5]. In both cases, we inject synthetic label noise into the pre-training datasets and later evaluate the resulting representations on downstream medical tasks.

**Evaluation.** To assess how pre-training label noise affects the transferability of learned representations in realistic clinical scenarios, we exclusively conduct out-of-domain (OOD) evaluation. Following [18], we evaluate the generalization capacity of the pre-trained feature extractor $f_\phi$ under varying levels of synthetic label noise. Given a downstream dataset $\mathcal{D}' = \{(x_i, y_i)\}_{i \in [M]}$, we freeze $f_\phi$ and train a $C$-way linear classifier on top of its extracted features using the standard linear probing (LP) protocol. The linear probing can be viewed as a simple black-box tuning method for pre-trained models that are typically large and difficult or unable to fully fine-tune.

**Experiment setup.** We use ImageNet-1K (IN-1K) [24] in fully supervised pre-training and YFCC15M [23] in CLIP pre-training, with ResNet-50 [25]. To simulate label noise during pre-training, we uniformly flip the ground truth class label into the other classes in IN-1K and randomly swap the text description from another image-text pair in YFCC15M. The noise ratio $\gamma$ is set to $\{0\%, 5\%, 10\%, 20\%, 30\%\}$, where $\gamma = 0\%$ corresponds to clean pre-training. For downstream evaluation, we focus on clinically relevant OOD tasks using Camelyon17, HAM10000, and NIHChestXray,

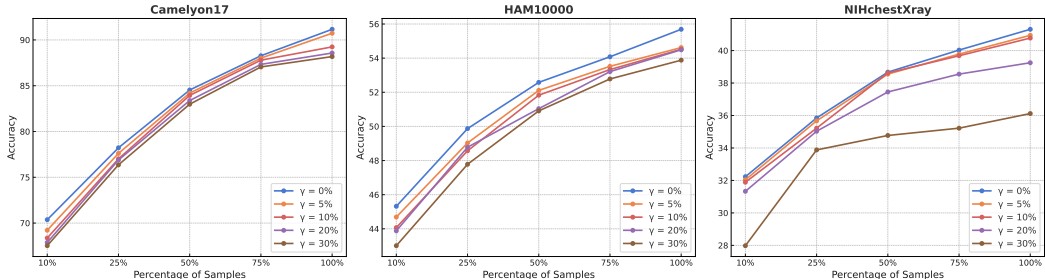

Figure 2: Average evaluation results of ImageNet-1K (IN-1K) fully supervised pre-training on downstream tasks with various percentages of data using ResNet-50. The robustness performance constantly decreases once noise is introduced in pre-training.

covering histopathology, dermatology, and thoracic radiology. We pre-train models on the clean and noisy variants of the above datasets, and then evaluate the standard test sets, which are fixed across all pre-training settings. We apply linear probing (LP) on top of frozen representations as a lightweight adaptation method and report accuracy on each OOD task. To ensure consistency, we use the same data split, augmentation, and training protocol across all settings. Additional implementation details are provided in AppendixB.

This choice of pretraining datasets reflects common practice in medical transfer learning. Due to the scarcity and privacy constraints of labeled medical data, foundation models used in clinical applications, such as PLIP [6] or MedCLIP [7], are typically pretrained on natural image–text corpora before being adapted to medical tasks. Even when medical-scale pretraining data exists (e.g., image–caption pairs scraped from social media), it is rarely open-sourced and often suffers from label ambiguity, weak alignment, or platform-specific bias. PLIP, for instance, is trained on over 200,000 pathology image–text pairs from Twitter, but the raw dataset is unavailable and difficult to verify.

## 2.2 Results: Noisy Pre-training Impairs OOD Performance in Medical Tasks

Figure 2 presents accuracy trends on Camelyon17, HAM10000, and NIH ChestX-ray14 using ResNet-50 pretrained on ImageNet-1K with varying noise levels. All models are evaluated via linear probing on frozen representations. Several consistent and task-specific patterns emerge. First, even mild pre-training noise (5% or 10%) leads to measurable degradation across all benchmarks, highlighting the sensitivity of medical transfer performance to upstream supervision quality. Second, the severity of degradation varies by task: Camelyon17 exhibits the sharpest decline, likely due to its fine-grained tissue structures and subtle class boundaries. In contrast, NIH ChestX-ray14 shows a flatter curve, possibly due to coarser visual categories and greater label redundancy. Third, the performance gap between clean and noisy pretraining widens as more downstream data is used, suggesting that early-stage noise imposes a ceiling that cannot be overcome by fine-tuning alone. These results suggest that label noise in pretraining introduces persistent structural damage in the learned features.

## 2.3 Feature Space and Logit Space Analysis

To understand how label noise in pre-training affects the structure of learned representations, we conduct an empirical analysis of higher-order statistical moments. Specifically, we compute skewness and kurtosis over feature embeddings and logit outputs on downstream medical datasets. These statistics provide a fine-grained view of the representational geometry, offering insights beyond spectral norms or singular values.

Take the feature space as an example (the same analysis applies to logit space). For each downstream dataset $\mathcal{D}' = \{(x_i, y_i)\}_{i \in [M]}$, we extract the feature matrix $F \in \mathbb{R}^{M \times D}$ from the *frozen* backbone $f_\phi$. We then compute the skewness and kurtosis of each feature dimension $j \in \{1, \ldots, D\}$ over all $M$ samples, where $F_{:,j}$ denotes the $j$-th feature dimension.

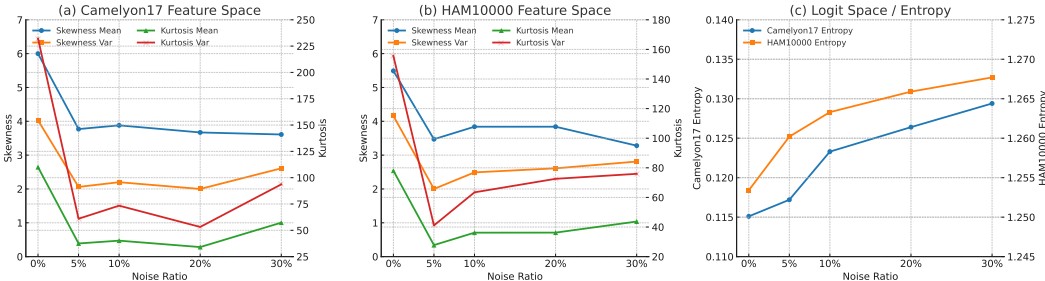

Figure 3: **Effect of pre-training label noise on representation statistics and prediction uncertainty.** (a) and (b): As the pre-training noise ratio increases, both skewness and kurtosis of the feature embeddings on Camelyon17 and HAM10000 decrease sharply, indicating a structural flattening of the learned feature space. This collapse is especially evident at low to moderate noise levels (5%–10%). (c): Logit entropy on downstream tasks increases monotonically with noise, suggesting higher predictive uncertainty and reduced confidence. Together, these trends highlight the degradation of representational quality and robustness due to label noise in pre-training.

**Definition 1** (Feature-wise Skewness). The skewness of feature dimension $j$ is defined as:

$$\text{Skew}(F_{:,j}) = \frac{M}{(M-1)(M-2)} \sum_{i=1}^{M} \left( \frac{F_{i,j} - \mu_j}{\sigma_j} \right)^3, \tag{1}$$

where $\mu_j = \frac{1}{M} \sum_{i=1}^{M} F_{i,j}$ and $\sigma_j = \sqrt{\frac{1}{M-1} \sum_{i=1}^{M} (F_{i,j} - \mu_j)^2}$ denote the mean and standard deviation of $F_{:,j}$, respectively.

**Definition 2** (Feature-wise Kurtosis). The kurtosis of feature dimension $j$ is defined as:

$$\text{Kurt}(F_{:,j}) = \frac{M(M+1)}{(M-1)(M-2)(M-3)} \sum_{i=1}^{M} \left( \frac{F_{i,j} - \mu_j}{\sigma_j} \right)^4 - \frac{3(M-1)^2}{(M-2)(M-3)}, \tag{2}$$

which quantifies the peakedness and tail heaviness of the distribution.

**Analysis.** We report the average skewness and kurtosis across feature dimensions to characterize the global shape of learned representations. As shown in Figure 3, both statistics consistently decline with increasing pre-training noise. This trend reflects a structural flattening of the feature space: representations become more symmetric and less peaked. It is indicative of reduced expressiveness and discriminability. In histopathology, where lesion patterns are compact and localized, clean pretraining naturally yields feature distributions with higher skewness and kurtosis. Noise disrupts this structure, degrading the model's ability to distinguish subtle visual cues.

We observe a similar trend in logit space: entropy increases monotonically with noise, suggesting greater predictive uncertainty. Together, these results provide a statistical explanation for the accuracy drop reported in Section 8. They further establish skewness and kurtosis as effective indicators of structural collapse under noisy pre-training. In the next section, we propose a fine-tuning strategy that explicitly restores these statistics to recover robustness.

## 3 Mitigating the Noise with Regularization on Distributional Shape

In this section, we introduce a simple and lightweight fine-tuning strategy that restores key distributional properties of the learned representations. As shown in Section 2.3, pre-training noise reduces the skewness and kurtosis of downstream feature distributions, indicating a collapse in asymmetry and peakedness. We hypothesize that preserving these higher-order moments during adaptation can improve generalization, particularly in medical tasks where subtle, localized patterns are critical. Unlike prior work that focuses on spectral properties such as singular value entropy [18], our method directly regularizes skewness and kurtosis relative to a clean reference, capturing distributional structure more relevant to clinical signals. Empirically, we find that while [18] improves accuracy on Camelyon17 by only 1.93%, our method achieves a 6.51% gain, highlighting its effectiveness in recovering fine-grained discriminative features in medical domains.

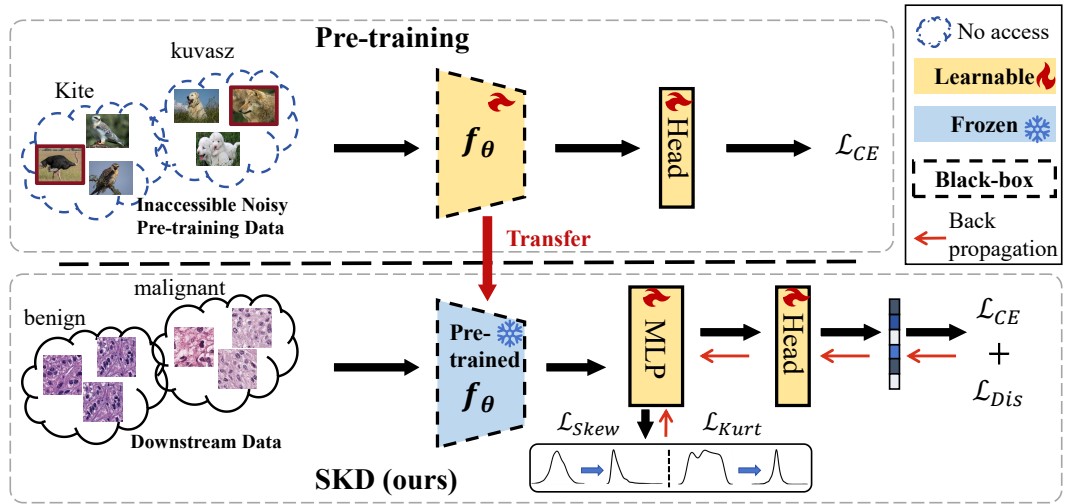

Figure 4: **Overview of the proposed framework.** In the pre-training stage (top), a backbone $f_\theta$ is trained on large-scale but inaccessible and noisy web data (e.g., ImageNet-1K or YFCC15M), resulting in potential label corruption. This pre-trained model is then transferred to downstream medical tasks (bottom), where only frozen features are accessible due to black-box constraints. Our method (SKD) introduces a lightweight MLP and classifier head on top of the frozen encoder. During fine-tuning, we apply three losses: (1) standard cross-entropy $\mathcal{L}_{CE}$, (2) skewness and kurtosis regularization $\mathcal{L}_{Skew}$, $\mathcal{L}_{Kurt}$ to restore feature asymmetry and peakedness, and (3) a disagreement loss $\mathcal{L}_{dis}$ to sharpen output margins.

## 3.1 Method

Let $f_\phi(x)$ denote the frozen pre-trained backbone and $g_\theta$ the classification head. As shown in Figure 4, we introduce a lightweight MLP between $f_\phi(x)$ and $g_\theta$, yielding transformed features $F \in \mathbb{R}^{B \times D}$ for a mini-batch of size $B$ and feature dimension $D$. These transformed features serve as the basis for both prediction and structural regularization.

**Skewness Regularization.** For each feature dimension $j \in \{1, \dots, D\}$, we compute the skewness of $F_{:,j}$ as:

$$\text{Skew}(F_{:,j}) = \frac{B}{(B-1)(B-2)} \sum_{i=1}^{B} \left( \frac{F_{i,j} - \mu_j}{\sigma_j} \right)^3, \tag{3}$$

where $\mu_j = \frac{1}{B} \sum_{i=1}^{B} F_{i,j}$ and $\sigma_j = \sqrt{\frac{1}{B-1} \sum_{i=1}^{B} (F_{i,j} - \mu_j)^2}$ denote the mean and standard deviation of the $j$-th feature dimension. The skewness loss penalizes deviation from a target skewness $\tau_s$:

$$\mathcal{L}_{\text{skew}} = \frac{1}{D} \sum_{j=1}^{D} \left| \text{Skew}(F_{:,j}) - \tau_s \right|. \tag{4}$$

**Kurtosis Regularization.** The kurtosis of each feature dimension is computed as:

$$\text{Kurt}(F_{:,j}) = \frac{B(B+1)}{(B-1)(B-2)(B-3)} \sum_{i=1}^{B} \left( \frac{F_{i,j} - \mu_j}{\sigma_j} \right)^4 - \frac{3(B-1)^2}{(B-2)(B-3)}, \tag{5}$$

and the corresponding loss is defined by deviation from a target $\tau_k$:

$$\mathcal{L}_{\text{kurt}} = \frac{1}{D} \sum_{j=1}^{D} \left| \text{Kurt}(F_{:,j}) - \tau_k \right|. \tag{6}$$

$\tau_s$ and $\tau_k$ are target statistics that can be computed from a clean model or empirically set to enforce non-degenerate representation structure.

**Disagreement Regularization.** Due to the low dimensionality of the logit space, computing skewness and kurtosis on $g_\theta(F)$ is unstable. Instead, we adopt a disagreement-based loss that enhances output separability by enlarging the margin between the ground-truth logit and the average of incorrect logits. Let $h(x) := g_\theta(F)$ be the logit output for input $x$, and let $y$ be the ground-truth class. We define:

$$\mathcal{L}_{\text{dis}}(x, y) = \frac{1}{\log 2} \log \left( 1 + \exp \left( h(x)_y - \frac{1}{|\mathcal{Y}| - 1} \sum_{\hat{y} \neq y} h(x)_{\hat{y}} \right) \right), \tag{7}$$

where $\mathcal{Y}$ is the label set and $h(x)_{\hat{y}}$ denotes the logit score for class $\hat{y} \neq y$.

**Overall Objective.** The final loss combines task supervision with the three regularization terms:

$$\mathcal{L} = \mathcal{L}_{\text{task}} + \lambda_s \mathcal{L}_{\text{skew}} + \lambda_k \mathcal{L}_{\text{kurt}} + \lambda_d \mathcal{L}_{\text{dis}}, \tag{8}$$

where $\lambda_s$, $\lambda_k$, and $\lambda_d$ are hyperparameters balancing each term. We set $\lambda_s = 0.1$, $\lambda_k = 1$, and $\lambda_d = 0.1$.

**Discussion.** This strategy explicitly preserves higher-order structural signals in the feature and output spaces, countering the flattening effect of noisy pre-training. It requires no changes to the pre-trained backbone and applies to frozen or partially fine-tuned settings, making it well-suited for adapting black-box foundation models in medical domains.

## 3.2 Evaluation on Noisy Medical Pre-training

We evaluate the effectiveness of our proposed SKD on noisy pre-trained models using downstream medical tasks. We compare our method against standard linear probing (LP) and NML [19].

We consider models pre-trained with various levels of synthetic label noise on ImageNet-1K and YFCC15M, and assess their performance on medical OOD benchmarks, as introduced in Section 2.1. In Table 1, we plot the average classification accuracy across these datasets. As in prior findings, LP performance drops steadily with increasing noise. While NML demonstrates benefits on general natural image tasks, its effectiveness does not transfer well to medical datasets. Due to the unique distributional properties of medical data, NML can even introduce adverse effects in certain tasks.

In contrast, incorporating our skewness and kurtosis regularization significantly boosts generalization performance, recovering performance close to or exceeding the clean model baseline. These results support our core claim: that representation collapse under pre-training noise is not solely a function of model size or task loss, but a structural issue in distributional shape. By explicitly regularizing skewness and kurtosis, we are able to reshape the learned features toward more expressive, asymmetric, and high-peaked distributions, resulting in improved OOD robustness across diverse medical domains.

## 4 Experiments

We further validate SKD on practical large-scale vision and language models that are pre-trained on noisy data, and discuss the noisy label learning and running time analysis in this section.

### 4.1 Vision Models and Datasets

**Setup.** To further evaluate our method in real-world medical scenarios, we conduct experiments on PLIP [6], a vision-language model pre-trained on over 2.3 billion noisy image-text pairs from the web. We use the four official datasets included in PLIP's release for downstream evaluation: Kather colon [27], PanNuke [28], DigestPath [29], and WSSS4LUAD [30]. These datasets cover diverse pathology classification tasks with varying levels of granularity. For comparison, we include the Zero-Shot and LP(origin) baselines reported in the original PLIP paper, as well as our own implementation of linear probing (LP), the recently proposed NML [18], and our method SKD.

**Results.** Table 2 reports the F1 and accuracy scores across all datasets. Our method (SKD) consistently achieves the best performance, surpassing both the baseline and NML across the board. Notably, SKD improves the F1 score from 0.931 (NML) to 0.959 on Kather colon, and from 0.948 to 0.956 on PanNuke. These gains are observed even when the original PLIP model achieves strong zero-shot performance, demonstrating that structural regularization remains beneficial during fine-tuning. The consistent improvements highlight the robustness of SKD against inherited noise in web-scale pre-training, particularly in pathology tasks that demand fine-grained discrimination.

Table 1: Classification accuracy under different pre-training noise ratios on three downstream medical dataset. SKD is our proposed method. NML is the previous SOTA. Bold indicates the best performance under each noise level.

| Pretrained | Dataset | Method | 0% | 5% | 10% | 20% | 30% | Avg Gain |
|---|---|---|---|---|---|---|---|---|
| CLIP | Camelyon17 | LP | 83.94 | 81.88 | 81.71 | 81.06 | 80.18 | - |
| | | GCE | 83.12 | 82.74 | 82.21 | 81.57 | 80.68 | 0.31 |
| | | NML | 80.61 | 84.21 | 83.32 | 84.68 | 85.61 | 1.93 |
| | | **SKD** | **89.48** | **89.59** | **89.50** | **86.08** | **86.65** | **6.51** |
| | HAM10000 | LP | 49.36 | 47.78 | 45.88 | 45.07 | 45.26 | - |
| | | GCE | 50.54 | 48.44 | 47.20 | 45.89 | 45.98 | 0.94 |
| | | NML | 50.76 | 51.35 | 49.03 | 52.78 | **50.04** | 4.12 |
| | | **SKD** | **55.19** | **55.88** | **55.75** | **53.32** | 49.84 | **7.33** |
| | ChestX-ray | LP | 44.75 | 42.00 | 42.78 | 41.58 | 41.75 | - |
| | | GCE | 45.42 | 42.71 | 43.11 | 42.06 | 42.13 | 0.51 |
| | | NML | 36.02 | 35.71 | 35.92 | 36.58 | 37.19 | -6.29 |
| | | **SKD** | **45.92** | **45.81** | **45.48** | **45.45** | **45.86** | **3.13** |
| ResNet50 | Camelyon17 | LP | 91.16 | 90.73 | 89.24 | 88.57 | 88.18 | - |
| | | GCE | 91.11 | 91.43 | 89.48 | 89.12 | 88.64 | 0.38 |
| | | NML | 89.27 | 92.44 | 88.09 | **90.51** | **91.29** | 0.74 |
| | | **SKD** | **91.76** | **92.50** | **91.39** | 89.12 | 89.02 | **1.18** |
| | HAM10000 | LP | 55.69 | 54.62 | 54.52 | 54.49 | 53.88 | - |
| | | GCE | 55.73 | 54.79 | 54.52 | 54.51 | 54.46 | 0.16 |
| | | NML | 54.71 | 55.16 | 54.21 | 54.77 | 50.67 | -0.74 |
| | | **SKD** | **58.25** | **57.54** | **56.94** | **58.37** | **57.54** | **3.09** |
| | ChestX-ray | LP | 41.31 | 35.94 | 40.77 | 39.25 | 36.12 | - |
| | | GCE | 41.37 | 36.28 | 41.23 | 40.02 | 36.67 | 0.44 |
| | | NML | 38.94 | 36.40 | 36.55 | 37.38 | 38.79 | -1.07 |
| | | **SKD** | **44.81** | **43.37** | **44.22** | **44.25** | **45.39** | **5.73** |

## 4.2 Language Models and Datasets

**Setup.** We evaluate the robustness of our method on 32 biomedical named entity recognition (NER) datasets spanning various subdomains, including disease, gene, and chemical recognition. Due to space constraints, we report representative results on five widely-used benchmarks in Table 3, with full results deferred to Appendix C. We fine-tune PubMedBERT [10] as the base model under three settings: (1) standard fine-tuning (Baseline), (2) noise-aware training using the NML framework, and (3) our proposed SKD method. All models are trained on clean downstream data, isolating the impact of noisy pre-training. Evaluation is conducted using both F1 score and accuracy metrics.

**Results.** As shown in Table 3, our method SKD consistently outperforms both the standard Baseline and NML across all five representative datasets, demonstrating superior robustness and discriminative capability under noisy pre-training. For example, on BC2GM, SKD achieves an F1 of 0.9459 and accuracy of 0.9501, improving significantly over Baseline (0.9053 / 0.9222). Notably, while NML improves over Baseline in some cases (e.g., BC4CHEMD), it fails to deliver consistent gains and even underperforms in others, likely due to the domain-specific challenges of biomedical NER. In contrast, SKD provides stable improvements by regularizing internal distributions, which helps preserve structural integrity even under distribution shift.

## 4.3 Additional Experiments on Large-Scale Foundation Models

To assess scalability beyond the backbones used in our main study, we further evaluate SKD on large-scale vision and language foundation models, aligning with the decision feedback.

**Vision.** We evaluate the proposed SKD regularization on a ViT-L [31] backbone pre-trained on ImageNet-21K with synthetic noisy labels. SKD consistently improves robustness across all medical benchmarks, achieving **93.4%** on Camelyon17, **60.0%** on HAM10000, and **45.9%** on NIH ChestXray,

Table 2: Real-world evaluation on PLIP using its original medical datasets. SKD consistently outperforms baselines across F1 and accuracy.

| Model | Dataset | Method | F1 | Accuracy |
|---|---|---|---|---|
| PLIP | Kather colon | Zero-Shot | 0.565 | - |
| | | LP(origin) | 0.877 | - |
| | | LP | 0.899 | 0.895 |
| | | NML | 0.931 | 0.929 |
| | | **SKD** | **0.959** | **0.959** |
| | PanNuke | Zero-Shot | 0.656 | - |
| | | LP(origin) | 0.902 | - |
| | | LP | 0.930 | 0.930 |
| | | NML | 0.948 | 0.948 |
| | | **SKD** | **0.956** | **0.956** |
| | DigestPath | Zero-Shot | 0.832 | - |
| | | LP(origin) | 0.856 | - |
| | | LP | 0.968 | 0.968 |
| | | NML | **0.979** | **0.979** |
| | | SKD | 0.976 | 0.969 |
| | WSSS4LUAD | Zero-Shot | 0.734 | - |
| | | LP(origin) | 0.927 | - |
| | | LP | 0.952 | 0.952 |
| | | NML | 0.956 | 0.956 |
| | | **SKD** | **0.958** | **0.958** |

Table 3: Real-world evaluation on biomedical NER tasks using PubMedBERT across five datasets. SKD consistently improves both F1 and accuracy over LP and NML, demonstrating its effectiveness beyond medical imaging.

| Dataset | Method | F1 | Accuracy |
|---|---|---|---|
| BC2GM | LP | 0.9053 | 0.9222 |
| | NML | 0.9187 | 0.9271 |
| | **SKD** | **0.9459** | **0.9501** |
| NCBI-disease-IOB | LP | 0.9330 | 0.9355 |
| | NML | 0.9378 | 0.9402 |
| | **SKD** | **0.9459** | **0.9471** |
| JNLPBA | LP | 0.8895 | 0.8825 |
| | NML | 0.9032 | 0.8957 |
| | **SKD** | **0.9211** | **0.9128** |
| BC4CHEMD | LP | 0.9373 | 0.9485 |
| | NML | 0.9506 | 0.9567 |
| | **SKD** | **0.9706** | **0.9725** |
| BioNLP11EPI-IOB | LP | 0.9286 | 0.9401 |
| | NML | 0.9362 | 0.9426 |
| | **SKD** | **0.9481** | **0.9492** |

outperforming LP, GCE, and NML in every case. Full results and training configurations are provided in Appendix D.1.1.

**Language.** Beyond PubMedBERT, we evaluate SKD on GPT-2[32] for biomedical NER (AnatEM, BC2GM, BC5CDR-chem, BC5CDR-disease, BioNLP09). SKD consistently matches or surpasses strong baselines, reaching **93.9%** (AnatEM), **91.8%** (BC2GM), and **95.7%** (BC5CDR-chem) with robust improvements over LP/GCE/NML. Detailed tables are in Appendix D.1.2.

## 4.4 Robustness under Structured Noise

In addition to uniform random noise, real-world medical data often exhibit structured or semantic label corruption (e.g., hierarchical mislabeling within taxonomies or co-occurrence-based confusion). To better reflect such conditions, we follow the reviewers' suggestion and construct a *taxonomy-aware hierarchical noise* setting.

We use ImageNet as the base taxonomy and synthetically replace a controlled percentage of image labels with their WordNet hypernyms (e.g., "Labrador Retriever" → "dog"), simulating realistic structured mislabeling commonly observed in annotation pipelines. We then train a ResNet-50 model from scratch under different hierarchical noise levels (0–30%) and compare four methods: *LP* (standard linear probing), *GCE* [33], *NML* (spectrum-preserving baseline), and our proposed *SKD*. All hyperparameters follow the random-noise setting for fair comparison.

As shown in Table D.2, SKD remains consistently robust under structured noise. On HAM10000, SKD improves over NML and GCE at every corruption level, e.g., **+1.3%** at 1% noise and **+0.4%** at 30%. This demonstrates that skewness–kurtosis regularization effectively mitigates representation collapse even when the label corruption is semantically structured. Full tables and visualization examples are provided in Appendix D.2.

## 4.5 In-depth Analysis

**Ablation study.** We conduct controlled ablations to isolate the effects of $\mathcal{L}_{skew}$, $\mathcal{L}_{kurt}$, and $\mathcal{L}_{dis}$ in SKD. Results on Camelyon17 show that removing $\mathcal{L}_{kurt}$ leads to the largest performance drop ($-1.07\%$ on average), followed by $\mathcal{L}_{skew}$ ($-0.47\%$) and $\mathcal{L}_{dis}$ ($-0.35\%$). These findings suggest that each term contributes to structural preservation, with kurtosis regularization playing a particularly important role. Full results are provided in Appendix E.

**Hyperparameter sensitivity.** We vary the weights of $\mathcal{L}_{\text{skew}}$, $\mathcal{L}_{\text{kurt}}$, and $\mathcal{L}_{\text{dis}}$ across several magnitudes and observe that SKD remains stable throughout. On Camelyon17, the largest performance fluctuation under weight variation is only $1.12\%$ for $\mathcal{L}_{\text{dis}}$, while $\mathcal{L}_{\text{skew}}$ varies by just $0.54\%$. These results indicate that SKD is robust to loss-weight choices, and a coarse grid search is sufficient.

In addition, we analyze the sensitivity of the target hyperparameters $\tau_s$ and $\tau_k$, which specify the desired skewness and kurtosis of the feature distribution. Varying $\tau_s$ and $\tau_k$ within $[-3, 3]$ on HAM10000 changes accuracy by less than $1\%$, indicating that SKD is insensitive to the exact target values. Hence, default settings ($\tau_s{=}0$, $\tau_k{=}0$) are sufficient for stable performance. Detailed results are provided in Appendix D.3.

## 5 Conclusion

This paper presents the first study of catastrophic inheritance in medical foundation models, where label noise in pretraining leads to structural collapse in learned representations and harms downstream robustness under distribution shift. Through controlled experiments across supervised and contrastive pretraining settings, we demonstrate that even mild pretraining noise causes consistent degradation in OOD generalization, particularly in high-resolution, fine-grained medical tasks. We trace this degradation to a flattening of feature and logit distributions, as reflected in lower skewness and kurtosis. To counter this effect, we propose a lightweight fine-tuning strategy that regularizes higher-order moments in both feature and output spaces. Our method introduces no architectural changes and applies to frozen or partially tunable backbones, making it compatible with real-world black-box medical models. Extensive results show that this approach restores representational sharpness and improves accuracy across diverse domains and modalities. Our findings highlight the importance of distribution-aware adaptation for safe deployment of foundation models in clinical settings.

## 6 Limitation

This work primarily investigates the effect of synthetic label noise in supervised and contrastive pretraining, focusing on its structural impact on downstream medical tasks. While our findings are consistent across multiple datasets and domains, the study is limited in two respects. First, all experiments are conducted using mid-sized models such as ResNet-50 and ViT-B/32 due to computational constraints. While these are representative, scaling to high-capacity architectures like ViT-L or LLaVA-Med remains unexplored. Second, the noise model assumes uniformly random corruption, which may underestimate the complexity of real-world noise such as co-occurrence bias or semantic drift. Extending the analysis to more realistic noise sources and larger model regimes will be important to fully characterize catastrophic inheritance in the wild.

### Acknowledgments and Disclosure of Funding

This work was supported in part by the National Natural Science Foundation of China under Grants U23A20389 and 62176139, and by the Qilu Young Scholars Program.

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

# A    Related Work

**Pretrain–Finetune Paradigm in the Medical Domain.** The pretrain–finetune (PT–FT) paradigm [1, 34, 35, 2] has become central to medical artificial intelligence, allowing models to leverage large-scale pretraining on general web-scale datasets followed by task-specific adaptation. This approach has been widely adopted in both medical imaging and biomedical natural language processing (NLP), where labeled data are often scarce and annotation costly [3, 36, 4]. In medical imaging, models such as PLIP [6, 37, 6] fine-tune CLIP [5] on pathology images and text, enabling effective zero-shot retrieval and classification. MedFILIP [38] incorporates domain-specific supervision into contrastive vision-language pretraining, while MedKLIP [39] enhances medical grounding through knowledge injection. MISS [40] formulates visual question answering as a generative task and achieves strong results with limited multimodal data. Other efforts [41, 42, 7] focus on adapting general-purpose vision-language models to medical domains under distribution shift. In biomedical NLP, pretraining on domain-specific corpora has led to a series of strong models: BioBERT [8], ClinicalBERT [9], and PubMedBERT [10], which all show clear improvements on downstream tasks such as disease mention recognition, drug–gene relation extraction, and clinical report parsing.

**Catastrophic Inheritance.** Chen et al. [18, 43] examine this effect of *catastrophic inheritance* through singular value decomposition, revealing that noise compresses the feature spectrum and diminishes the capacity of principal components. To mitigate this, they introduce NMTune, which enhances robustness by enforcing spectral preservation during fine-tuning. Building on similar insights, Chang et al. [44] propose BA-LoRA for large language models, combining low-rank adaptation with spectral regularization. Separately, Li et al. [45] investigate how generative models inherit structural biases from pretraining distributions, with a focus on text generation tasks.

# B    Understanding The Noisy Labels In Pre-Training Data

We provide additional experiment details for the motivating example of ResNet-50 in this section. We also present the detailed results on each downstream dataset for noisy pre-trained models on both ImageNet-1K and YFCC15M.

## B.1    Pre-training datasets and Hyper-parameters

For analysis in Section 2, we conduct pre-training of ResNet-50 on ImageNet-1K and YFCC15M. For ImageNet-1K pre-training, we follow the training recipe in Wightman et al. (2021). To introduce noise in ImageNet-1K, we use function cleanlab (Northcutt et al., 2021) to introduce symmetric noise in each class. For YFCC15M CLIP pre-training, we follow the training recipe in Cherti et al. (2023). To introduce noise in YFCC15M, we swap the text description between two randomly sampled image-text pairs until the noise ratio is achieved. We show the validation accuracy on ImageNet-1K of the noisy ResNet-50 models pre-trained on ImageNet-1K and zero-shot accuracy on ImageNet-1K of the noisy ResNet-50 models pre-trained on YFCC15M in Table 3. The results show that our pre-training achieves the state-of-the-art results (Wightman et al., 2021; Cherti et al., 2023), as a basis for our further analysis.

## B.2    Downstream Vision Datasets and Hyper-parameters

We present the details of the in-domain (ID) vision datasets in Table 4 and out-of-domain vision datasets Table 5. For ID, we conduct training on the training set and test on the validation set of the downstream dataset. For OOD on DomainNet (Peng et al., 2019), we conduct training on the training set of DomainNet Real or DomainNet Sketch, and test on all the other three DomainNet datasets not used in training. For OOD on ImageNet (Russakovsky et al., 2015), we conduct training on ImageNet training split and test on its variants. To transfer a pre-trained model, we use linear probing (LP) for analysis as shown in Section 2. We train the linear classifier for 30 epochs on each downstream dataset, using AdamW (Kingma Ba, 2014) optimizer with a cosine scheduler. We do not use weight decay for linear probing and set the learning rate to 0.1 for all tasks.

Table 4: ImageNet-1K validation and zero-shot accuracy of ImageNet-1K pre-trained and YFCC15M CLIP pre-trained noisy ResNet-50 models.

| Noise Ratio | ImageNet-1K Pre-train Validation Accuracy | YFCC15M CLIP Pre-train Zero-shot Accuracy |
|:---:|:---:|:---:|
| 0% | 79.96 | 32.64 |
| 5% | 79.18 | 30.86 |
| 10% | 78.61 | 29.54 |
| 20% | 76.27 | 27.72 |
| 30% | 73.11 | 26.53 |

Table 5: Details of the 3 medical vision datasets used to evaluate transfer performance.

| Dataset | Classes | Train Size | Test Size | Evaluation Metric |
|:---|:---:|:---:|:---:|:---:|
| Camelyon17 | 2 | 302,436 | 33,501 | accuracy |
| HAM10000 | 7 | 8,138 | 2,000 | accuracy |
| NIH ChestX-ray | 14 | 89,322 | 25,596 | accuracy |

## B.3 Detailed results

Figure 5 reports the detailed accuracy trends across different levels of synthetic label noise for YFCC15M and ResNet-50 pre-training. Across all settings, SKD demonstrates consistent robustness compared to LP and NML. On Camelyon17, accuracy under YFCC15M pre-training (Figure 5a) remains above 83% up to 30% noise, with SKD maintaining a clear margin over both baselines. For HAM10000 (Figure 5b), the performance is more sensitive to noise, but SKD still preserves ~2–3% improvement under high-noise regimes. A similar trend is observed on ChestX-ray (Figure 5c), where accuracy degrades steadily, but SKD slows the collapse. Under supervised ResNet-50 pre-training (Figures 5d–f), SKD consistently achieves top performance across all datasets. For example, on Camelyon17 (Figure 5d), accuracy stays above 90% even at 30% label noise, whereas LP and NML drop significantly. These trends confirm the effectiveness of SKD in preserving feature integrity under both contrastive and supervised pre-training paradigms, particularly in safety-critical medical tasks.

## B.4 Detailed Feature and Logit Results

**Skewness and kurtosis degradation under noise.** We analyze the feature distributional changes of ResNet-50 models pre-trained with varying noise ratios by reporting the mean and variance of skewness and kurtosis on Camelyon17, HAM10000, and NIH ChestX-ray datasets. As shown in Figure 6, both skewness and kurtosis consistently degrade with increasing label noise. On Camelyon17, the skewness mean drops from 6.00 to 3.61, and kurtosis mean plummets from 110.02 to 57.28 as noise increases from 0% to 30%, accompanied by a sharp decline in kurtosis variance from 232.15 to 93.66. Similar trends are observed on HAM10000 (skewness mean: $5.49 \rightarrow 3.28$; kurtosis mean: $78.05 \rightarrow 43.71$) and NIH ChestXray (skewness mean: $5.11 \rightarrow 3.12$; kurtosis mean: $73.44 \rightarrow 24.14$). These shifts indicate a progressive flattening of the representation space under noise—feature dimensions become more symmetric (lower skewness) and less heavy-tailed (lower kurtosis), pointing to a collapse of expressive capacity. Such structural degradation motivates our SKD design, which explicitly regularizes these higher-order moments to preserve discriminative geometry.

**Logit-level analysis.** We examine how pre-training noise affects the distribution of logit outputs across different datasets. As shown in Figure 7, increasing noise levels lead to a consistent rise in logit entropy (`entropy_mean`), reflecting increased prediction uncertainty. For example, in NIHChestXray, entropy increases from 1.5710 at 0% noise to 1.6830 at 30% noise. Simultaneously, both logit energy (`energy_mean`) and maximum softmax probability (`msp_mean`) decrease, indicating lower confidence and greater dispersion in the predictions. On Camelyon17, energy drops from 5.1151 to 4.2888, while `msp_mean` falls from 0.9301 to 0.9006.

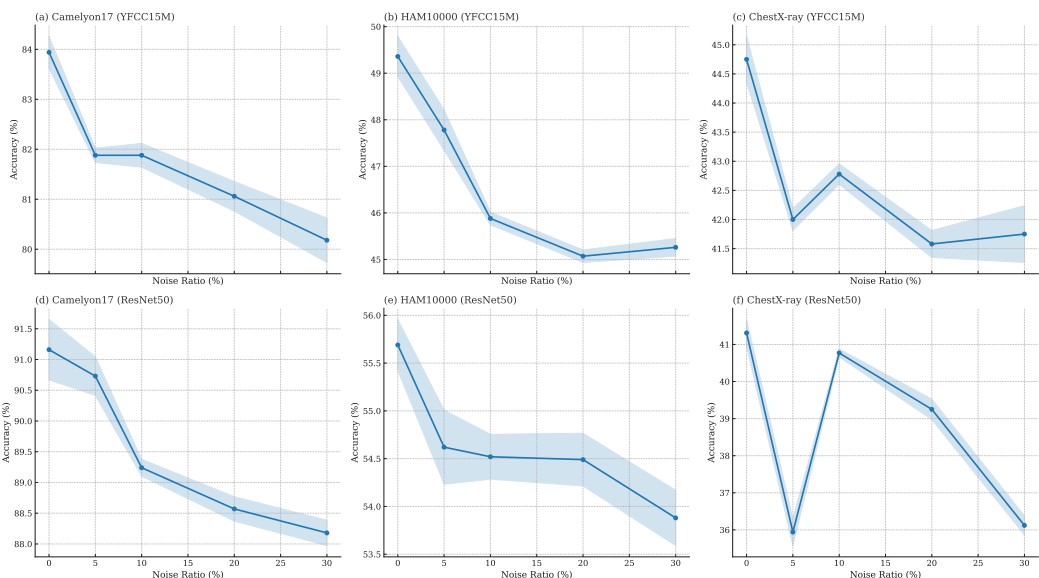

Figure 5: Average evaluation results of ImageNet-1K (IN-1K) fully supervised pre-training on downstream tasks with various percentages of data using ResNet-50. The robustness performance constantly decreases once noise is introduced in pre-training.

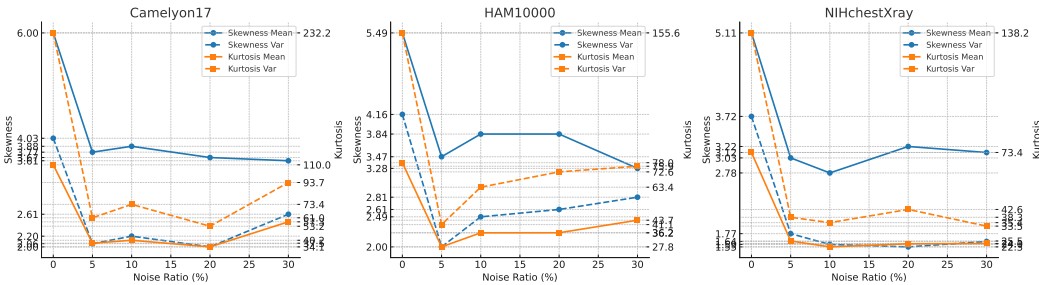

Figure 6: **Degradation of skewness and kurtosis under label noise.** We visualize the mean and variance of feature-wise skewness and kurtosis for ResNet-50 pre-trained with different levels of label noise. All three datasets (Camelyon17, HAM10000, NIH ChestXray) exhibit a consistent downward trend in both metrics as noise increases, indicating a flattening of the representation space. Skewness becomes closer to zero (more symmetric), and kurtosis drops significantly (less peaky), reflecting reduced expressiveness and discriminative structure.

These patterns are tightly connected to the structural degradation observed in feature-level statistics—specifically, reduced skewness and kurtosis under noisy supervision. Lower skewness suggests more symmetric and less distinctive feature distributions, whereas reduced kurtosis indicates a lack of peakedness and diminished confidence concentration. Together, these shifts in the representation space translate into softer and more ambiguous logit-level predictions, underscoring the downstream impact of representational collapse. Our results support the view that structural noise inherited during pre-training propagates through to the output layer, impairing model reliability in high-stakes clinical settings.

# C Experiment

More details of experiments in Section 4 are shown here.

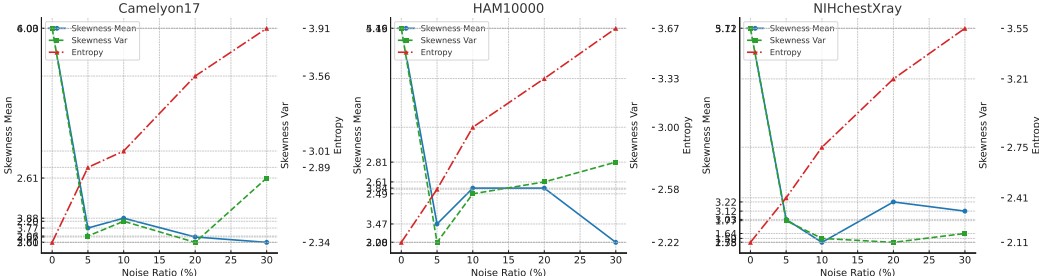

Figure 7: **Logit-level changes under pre-training noise.** We report logit entropy, energy, and maximum softmax probability (MSP) for CLIP models pre-trained with different noise levels on Camelyon17, HAM10000, and NIH ChestXray. As noise increases, entropy rises while both energy and MSP decrease, indicating higher prediction uncertainty and lower confidence. These trends mirror the flattening of feature-level skewness and kurtosis, suggesting that representational degradation propagates through to the output space.

Table 6: Details of the biomedical NER datasets used for evaluation. Each dataset provides token-level annotations for domain-specific entity types (e.g., diseases, chemicals, genes). Following prior work, we report both F1 score and accuracy.

| Dataset | Entity Type | Annotation Scheme | Evaluation Metric |
|---|---|---|---|
| BC2GM | Gene/protein | IOB | F1, Accuracy |
| BC4CHEMD | Chemicals | IOB | F1, Accuracy |
| CRAFT | Multiple (e.g., cell, gene) | IOB | F1, Accuracy |
| BC5CDR-chem | Chemicals | IOB | F1, Accuracy |
| BC5CDR-disease | Diseases | IOB | F1, Accuracy |
| JNLPBA | Biomedical terms | IOB | F1, Accuracy |
| NCBI-disease | Diseases | IOB | F1, Accuracy |
| BioNLP11 | Multiple event/mention types | IOB | F1, Accuracy |
| BioNLP13 | Multiple entity types (species, cell, etc.) | IOB | F1, Accuracy |
| Ex-PTM | Protein post-translational modifications | IOB | F1, Accuracy |
| AnatEM | Anatomical entities | IOB | F1, Accuracy |

## C.1 Detailed Setup For Language Model Experiment

To assess the generalizability of our method beyond vision, we evaluate its effectiveness in natural language processing (NLP) via biomedical named entity recognition (NER). Specifically, we use the `BiomedBERT` model as our encoder backbone. This model is a domain-adapted variant of BERT, pre-trained on PubMed abstracts and PMC full-text articles, making it well-suited for biomedical language tasks.

We conduct experiments across 32 standard biomedical NER benchmarks, including BC2GM, BC4CHEMD, NCBI-disease, JNLPBA, and multiple BioNLP and CRAFT datasets. These datasets cover a wide range of biomedical entity types such as genes, proteins, chemicals, diseases, and anatomical structures. A summary of representative datasets is provided in Table 6, while full results across all benchmarks are included in the supplementary materials.

We compare our proposed SKD method against linear probing (Baseline) and a recent baseline NML, reporting both F1 score and accuracy. Results demonstrate that SKD consistently improves upon prior methods across nearly all datasets, highlighting its robustness and transferability in language settings.

## C.2 Biomedical NER Results.

We evaluate the effectiveness of our method SKD on 32 biomedical named entity recognition (NER) datasets spanning various entity types (e.g., chemical, gene/protein, species, disease) from benchmark corpora such as BioNLP, CRAFT, BC2GM, and NCBI-disease. All experiments use the BioMedNLP-

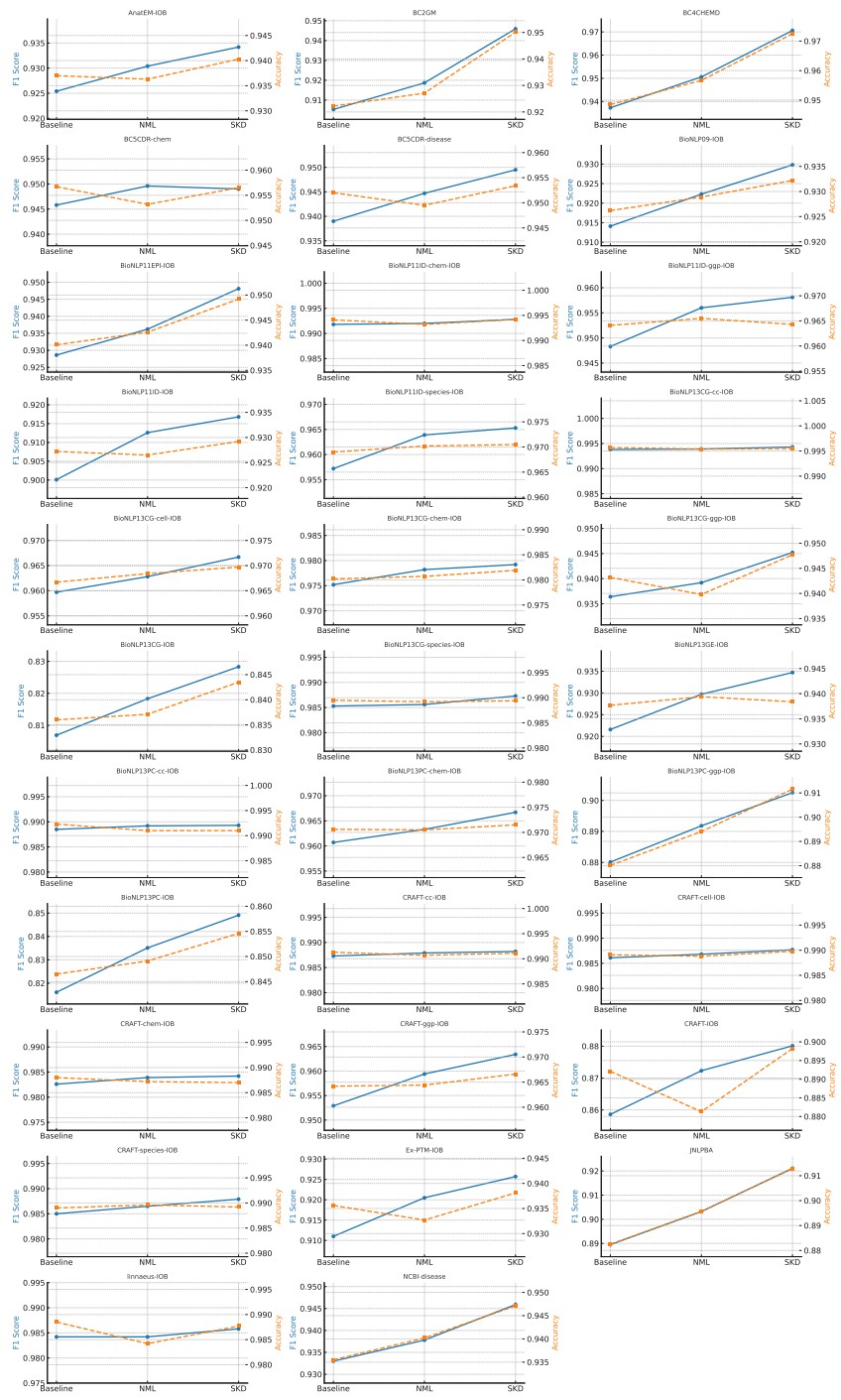

Figure 8: **F1 and Accuracy on Biomedical NER Benchmarks.** We report F1 score and accuracy for 32 biomedical NER datasets using the BioMedBERT backbone under three fine-tuning strategies: linear probing (LP), NML, and our proposed SKD.

BioMedBERT-base-uncased-abstract-fulltext model as the backbone, with fine-tuning conducted using three strategies: LP (Baseline), NML, and our proposed SKD.

As shown in Figure 8, SKD achieves consistent improvements across both F1 and accuracy metrics. For instance, on BC2GM, SKD achieves an F1 score of 0.9459 compared to 0.9053 (LP) and

0.9187 (NML); on BC4CHEMD, SKD further improves the F1 to 0.9706. Across all datasets, SKD outperforms prior methods by an average margin of **1.21**% in F1 and **0.57**% in accuracy. These results highlight SKD's ability to enhance robustness and representation quality in biomedical NLP tasks under noisy transfer settings.

# D  Extended Results and Analyses

## D.1  Large-Scale Foundation Models

### D.1.1  Vision: ViT-L (ImageNet-21K pretrain with noisy labels)

We extend the PLIP setting to a large-scale vision backbone, ViT-L [31], pre-trained on ImageNet-21K with noisy labels. Across three medical benchmarks, SKD consistently surpasses LP/GCE/NML, confirming that skewness–kurtosis regularization scales to large backbones.

Table 7: ViT-L under noisy pre-training. Best in **bold**.

| Model | Dataset | LP | GCE | NML | **SKD** |
|-------|---------|------|------|------|---------|
| ViT-L | Camelyon17 | 91.6 | 92.3 | 91.8 | **93.4** |
|       | HAM10000 | 58.8 | 59.3 | 58.6 | **60.0** |
|       | NIHChestXray | 43.5 | 44.3 | 43.8 | **45.9** |

### D.1.2  Language: GPT-2 for Biomedical NER

Beyond PubMedBERT, we evaluate SKD on GPT-2 [32] across five representative NER datasets. SKD matches or surpasses strong baselines on all benchmarks.

Table 8: GPT-2 on biomedical NER. Best in **bold**.

| Model | Dataset | LP | GCE | NML | **SKD** |
|-------|---------|------|------|------|---------|
| GPT-2 | AnatEM | 93.2 | 93.5 | 93.6 | **93.9** |
|       | BC2GM | 91.3 | 91.6 | 91.5 | **91.8** |
|       | BC5CDR-chem | 95.4 | 95.5 | 95.5 | **95.7** |
|       | BC5CDR-disease | 95.2 | 95.2 | 95.3 | **95.4** |
|       | BioNLP09 | 91.6 | 91.8 | 91.8 | **92.2** |

## D.2  Robustness under Structured (Hierarchical) Noise

We simulate taxonomy-aware hierarchical mislabeling by replacing labels with WordNet hypernyms at controlled ratios (e.g., "Labrador Retriever"→"dog"). SKD remains robust across all corruption levels on HAM10000.

Table 9: Taxonomy-aware hierarchical noise on HAM10000.

| Dataset | Noise(%) | LP | GCE | NML | **SKD** |
|---------|----------|------|------|------|---------|
| HAM10000 | 0 | 56.9 | 56.6 | 57.4 | **58.4** |
|          | 1 | 56.5 | 56.7 | 57.0 | **58.4** |
|          | 2 | 56.2 | 56.4 | 56.6 | **57.7** |
|          | 5 | 56.0 | 55.9 | 56.0 | **57.3** |
|          | 10 | 55.5 | 55.6 | 55.4 | **56.4** |
|          | 20 | 55.1 | 55.1 | 55.2 | **56.0** |
|          | 30 | 54.2 | 54.1 | 54.3 | **55.8** |

### D.3 Sensitivity Analyses

#### D.3.1 Target Statistics $(\tau_s, \tau_k)$

To evaluate the influence of the two *target* hyperparameters $\tau_s$ and $\tau_k$ (which specify the desired skewness and kurtosis), we conduct a detailed sensitivity analysis on the HAM10000 dataset. These targets were originally set to the negative values of the skewness and kurtosis computed from a clean model, but we also explore a range of alternative settings to assess robustness.

The results in Table 10 show that SKD is largely **insensitive** to the exact choice of $(\tau_s, \tau_k)$. This is expected since the primary role of SKD is to encourage non-degenerate skewness and kurtosis, counteracting representation flattening rather than matching precise target values. Across a wide range of target settings, the performance variation is within $1\%$, confirming that SKD remains stable even when the target statistics are misestimated. In all main experiments, we therefore simply fix $\tau_s = 0$ and $\tau_k = 0$ for consistency.

Table 10: Sensitivity of SKD to target $(\tau_s, \tau_k)$ on HAM10000 (Accuracy %).

| **HAM10000** | $\tau_s$ | $-3.0$ | $-1.0$ | $0.0$ | $1.0$ | $3.0$ |
|---|---|---|---|---|---|---|
| Accuracy | | 56.9 | 56.9 | 56.9 | 56.9 | 56.8 |
| **HAM10000** | $\tau_k$ | $-30.0$ | $-10.0$ | $0.0$ | $10.0$ | $30.0$ |
| Accuracy | | 56.9 | 56.8 | 56.9 | 56.8 | 56.6 |

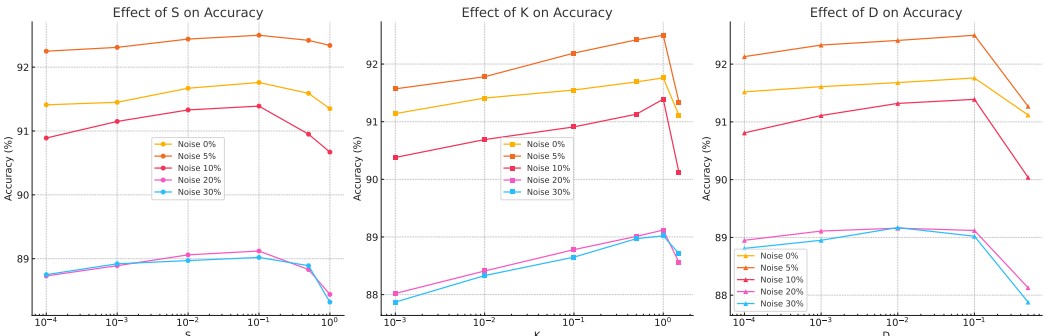

Figure 9: **Hyperparameter sensitivity of SKD components on Camelyon17.**

#### D.3.2 Loss Weights $(\lambda_{\textbf{skew}}, \lambda_{\textbf{kurt}}, \lambda_{\textbf{dis}})$

On Camelyon17, varying loss weights across orders of magnitude gives at most $1.12\%$ fluctuation (for $\mathcal{L}_{\text{dis}}$), while $\mathcal{L}_{\text{skew}}$ varies by only $0.54\%$. See Fig. 9 for full curves.

### E Ablation study

We conduct ablation experiments to disentangle the contributions of the three components in SKD: skewness regularization (S), kurtosis regularization (K), and logit disagreement regularization (D). As shown in Table 11, removing any individual component consistently reduces performance across all noise levels and datasets. Among the single-component variants, kurtosis (K) alone typically performs best, aligning with our hypothesis that high-order statistics play a dominant role in mitigating representation collapse. Combinations of two terms improve upon single terms, while the full SKD achieves the best average performance on Camelyon17 (90.76%), HAM10000 (57.73%), and NIHChestXray (44.41%), highlighting the complementary effects of the three regularizers.

Table 11: **Ablation study on SKD components using ResNet-50.** We evaluate variants of SKD by selectively removing components: skewness (S), kurtosis (K), and disagreement (D), across three medical datasets. The full SKD consistently achieves the best performance under varying pre-training noise ratios, confirming the complementary benefits of each regularizer.

| Dataset | Method | 0 | 5 | 10 | 20 | 30 | Avg |
|---|---|---|---|---|---|---|---|
| **Camelyon17** | S | 90.10 | 91.05 | 88.24 | 88.77 | 88.10 | 89.25 |
| | K | 91.41 | 92.53 | 88.70 | 90.57 | 88.06 | 90.25 |
| | D | 91.02 | 90.77 | 89.01 | 88.27 | 88.78 | 89.57 |
| | S&K | 91.52 | 92.11 | 90.72 | 88.92 | 88.79 | 90.41 |
| | S&D | 91.09 | 91.54 | 90.22 | 87.88 | 87.74 | 89.69 |
| | K&D | 91.28 | 92.13 | 90.72 | 88.64 | 88.69 | 90.29 |
| | **SKD** | **91.76** | **92.50** | **91.39** | **89.12** | **89.02** | **90.76** |
| **HAM10000** | S | 56.02 | 55.25 | 55.55 | 57.53 | 55.72 | 56.01 |
| | K | 56.14 | 55.21 | 54.74 | 57.03 | 56.08 | 55.84 |
| | D | 55.95 | 53.73 | 55.62 | 56.45 | 55.17 | 55.38 |
| | S&K | 57.34 | 55.98 | 55.42 | 57.90 | 56.87 | 56.70 |
| | S&D | 56.87 | 56.45 | 55.68 | 57.44 | 56.15 | 56.52 |
| | K&D | 56.96 | 56.05 | 55.75 | 57.37 | 56.54 | 56.53 |
| | **SKD** | **58.25** | **57.54** | **56.94** | **58.37** | **57.54** | **57.73** |
| **NIHchestXray** | S | 41.75 | 39.90 | 40.66 | 41.20 | 41.91 | 41.08 |
| | K | 43.85 | 41.98 | 42.52 | 43.18 | 43.45 | 43.00 |
| | D | 41.70 | 38.56 | 41.02 | 40.35 | 38.23 | 39.97 |
| | S&K | 43.80 | 41.90 | 42.56 | 43.23 | 43.45 | 42.99 |
| | S&D | 43.57 | 42.08 | 42.24 | 42.89 | 43.58 | 42.87 |
| | K&D | 43.62 | 42.44 | 42.36 | 43.15 | 44.38 | 43.19 |
| | **SKD** | **44.81** | **43.37** | **44.22** | **44.25** | **45.39** | **44.41** |

Table 12: **Training time comparison on PanNuke using PLIP.**

| Dataset | LP | NML | SKD |
|---|---|---|---|
| PanNuke | 20(s) | 100(s) | 150(s) |

# F   Running Time

To assess computational efficiency, we measure the wall-clock training time of LP, NML, and SKD on the PanNuke dataset using the PLIP model. As shown in Table 12, LP is the fastest with only 20 seconds per run, while NML takes approximately 100 seconds. SKD introduces additional overhead due to the skewness, kurtosis, and disagreement regularization losses, requiring 150 seconds. Despite the added complexity, SKD remains lightweight and practical for real-world deployment, with only a $2.5\times$ increase over LP and a $1.5\times$ increase over NML. All experiments were conducted on a single NVIDIA A100 40GB GPU.

# G   More Discussion

## G.1   Future Work

We outline several future directions. One is to evaluate the proposed skewness–kurtosis regularization at scale, particularly on larger backbones and long-tailed medical tasks with high inter-class similarity. Another direction is to generalize the regularization beyond fixed targets: current methods use global statistics as anchors, but instance- or task-adaptive constraints may better preserve local geometry and class separability. Additionally, while our method focuses on preserving internal structure, integrating it with uncertainty estimation or confidence calibration could further improve reliability, especially in settings with weak supervision or low-resource deployment.

### G.2 Broader Impact

This study contributes to improving the robustness of foundation models in safety-critical domains such as healthcare. By addressing structural degradation caused by noisy pretraining, our method may reduce silent failures and improve model interpretability under distribution shift. However, as with any representation regularization method, care must be taken to ensure that the learned structure does not inadvertently suppress rare or underrepresented patterns. In high-stakes deployment, we recommend incorporating fairness audits, uncertainty quantification, and human oversight to prevent overconfidence and preserve trustworthiness.

