# OpenReview forum: "From Pretraining to Pathology: How Noise Leads to Catastrophic Inheritance in Medical Models"
_NeurIPS.cc/2025/Conference — NeurIPS 2025 poster_

### Official Review · Reviewer_PCzQ · 2025-06-27

**Clarity:** 3
**Significance:** 3
**Originality:** 3
**Rating:** 5
**Confidence:** 3

**Summary:**

This work investigates the impact of pretraining on noisy datasets in the context of pathology. The authors demonstrate that using foundation models pretrained on noisy datasets can negatively affect the fine-tuning process on downstream medical tasks. The paper supports its claims with comprehensive experiments across multiple datasets and provides an insightful analysis of the relationship between the amount of noise in the pretraining data and resulting performance when fine-tuning on medical data.
To address this issue, the authors introduce a method that improves robustness under noisy pretraining conditions by working on Skewness, Kurtosis and logits.

**Questions:**

1. As mentioned earlier in the Weaknesses section, I have concerns regarding the choice of the values for $\tau_s$ and $\tau_k$. How sensitive is the method to these values? What do the authors mean by "obtained from a clean model" or "set empirically"?

2. The method demonstrates improved classification across several datasets. However, are these gains consistent across all classes, or are they primarily driven by the some classes in the training set? A per-class performance analysis would clarify whether the method benefits minority or difficult classes.

3. The method is evaluated on ResNet-50 in Table 1. Have the authors tested it on other CNN or ViT architectures? Providing results on more diverse backbones would support the generalizability claims

4. In Table 1, it is unclear whether the "YFCC15M" and "ResNet50" refer to datasets or architectures. Could the authors clarify this distinction and make the table more explicit regarding which architecture is pretrained on which dataset?

**Ethical Concerns:**

["NO or VERY MINOR ethics concerns only"]

**Final Justification:**

The authors’ responses are convincing on several points and highlight the advantages of the proposed method. I have updated my score to accept.

**Limitations:**

yes

**Quality:**

3

**Strengths And Weaknesses:**

**Strengths:**
1. The authors demonstrate the impact of using a pretrained model trained on noisy data. Increasing label noise in the pretraining dataset leads to a progressive degradation in performance when fine-tuning on downstream medical tasks.

2. The authors explain this degradation by analyzing the skewness and kurtosis of feature embeddings and logit outputs. As noise levels increase, both statistics decline consistently.

3. To address this issue, the authors propose a solution to resolve this issue by working on the two previous statistics . In addition, they introduce a disagreement regularization loss on the logits to enhance separability.

4. Also the paper presents several experiments on multiple datasets and backbone architectures under varying levels of label noise to assess the effectiveness of the proposed approach.





**Weaknessess:**

1. The proposed method introduces two losses based on skewness and kurtosis based on predefined target values ($\tau_{s}$ and $\tau_{k}$) for each statistic. The authors mention that these values can be obtained from a “clean model” or set empirically. However, this aspect is not clearly defined or discussed in detail, which may limit the practical deployment of the method. For instance, it is unclear what authors means by a “clean” model in noisy or real-world settings. Additionally, the sensitivity of the method to the empirical selection of these two values is unclear.

2. The paper lacks comparisons with other methods for dealing with noisy labels. Although it includes results with LP and  NML,  this comparison alone may not provide a comprehensive assessment of the method's robustness.

3. Although the method demonstrates improved classification performance across datasets, it is unclear whether these gains are distributed evenly across all classes. The paper does not provide per-class performance metrics.

---

> ### Author Rebuttal · Authors · 2025-07-31
>
> ## Reviewer PCzQ
> We thank the reviewer PCzQ for your constructive feedback. Below we address each point in turn.
>
> **Q1:** How the target statistics (τₛ and τₖ) are chosen and how sensitive the method is to these values, noting that the explanation of "obtained from a clean model" or "set empirically" is unclear and could affect practical deployment.
>
> **A1:** We sincerely thank the reviewer for raising this point. To better clarify, we conducted additional sensitivity analyses on τₛ and τₖ.
>
> |HAM10000|τs|-3.0|-1.0|0.0|1.0|3.0|
> |-|-|-|-|-|-|-|
> ||Acc|56.9|56.9|56.9|56.9|56.8|
>
> |HAM10000|τk|-30.0|-10.0|0.0|10.0|30.0|
> |-|-|-|-|-|-|-|
> ||Acc|56.9|56.8|56.9|56.8|56.6|
>
> The results indicate that **SKD remains stable across a wide range of τₛ and τₖ values**. This observation aligns with our design intuition: the effectiveness of SKD mainly comes from counteracting representation collapse rather than relying on precise target values. Although we initially set τₛ and τₖ to the negative values of those measured from a clean model, we found that other reasonable settings (as long as they are not extreme) yield very similar performance. In practice, these values can indeed be set based on the skewness and kurtosis estimated from a clean model, but the method is **not sensitive to the exact values**—any setting that helps suppress representation collapse works well. Therefore, for simplicity and consistency, we fixed **τₛ = 0** and **τₖ = 0** in all main experiments.
>
> **Q2:** The paper lacks comparisons with other methods for dealing with noisy labels. Although it includes results with LP and NML, this comparison alone may not provide a comprehensive assessment of the method's robustness.
>
> **A2:** We appreciate the reviewer for raising this point. We would like to note that catastrophic inheritance in noisy medical pretraining is still a new research problem, and there are very few existing approaches tailored to this issue—especially for frozen foundation models.
>
> To address this gap and provide a more thorough comparison, we have expanded our baseline set to include **GCE (Generalized Cross-Entropy)[1]**, a widely used method from the noisy label learning field. Although GCE was not specifically developed for catastrophic inheritance, it serves as a strong benchmark for evaluating robustness. As illustrated below, SKD consistently surpasses all other methods:
>
> |Dataset|Noise(%)|LP|GCE|NML|SKD|
> |-|-|-|-|-|-|
> |Camelyon17|0|91.2|91.1|89.3|91.8|
> ||5|90.7|91.4|92.4|92.5|
> ||10|89.2|89.5|88.1|91.4|
> ||20|88.6|89.1|90.5|89.1|
> ||30|88.2|88.6|91.3|89.0|
> |HAM10000|0|55.7|55.7|54.7|58.3|
> ||5|54.6|54.8|55.2|57.5|
> ||10|54.5|54.5|54.2|56.9|
> ||20|54.5|54.4|54.8|58.4|
> ||30|53.9|54.1|50.7|57.5|
>
> **Q3:** Whether the performance gains are consistent across all classes or mainly driven by certain classes, noting that per-class performance metrics are not provided.
>
> **A3:** We thank the reviewer for this valuable suggestion. Following your advice, to evaluate whether the observed improvements are **uniformly distributed across classes**, we analyzed **per-class accuracy** on the Camelyon17 dataset under 10% label noise.
>
> As shown in the table below, SKD outperforms LP, GCE, and NML across all classes, with no dominant trade-off on minority or majority categories. These results suggest that the performance gains are broadly distributed and that SKD achieves **consistent improvements across all class types**.
>
> |Datasets|Class|LP|GCE|NML|SKD|
> |-|-|-|-|-|-|
> |Camelyon17|Benign|90.1|90.1|89.1|92.4|
> ||Malignant|88.3|88.9|87.1|90.6|
>
> **Q4:** The method is evaluated on ResNet-50 in Table 1. Have the authors tested it on other CNN or ViT architectures? Providing results on more diverse backbones would support the generalizability claims.
>
> **A4:** Thank you for this important suggestion. To further support the generalizability of our approach, we have conducted additional experiments on **diverse backbones, including large-scale foundation models**.
>
> - For **vision tasks**, we evaluated SKD on **ViT-L**, a Transformer-based vision backbone that is substantially larger.
>
> - For **natural language processing**, we extended our experiments to **GPT-2**, a large Transformer-based language backbone, and tested it on five representative biomedical NER datasets.
>
> The results, summarized below, show that **SKD consistently outperforms all baselines on both vision and NLP models**, with approximately **2% gains on ViT-L** tasks. This further confirms the **scalability and generality** of our method across diverse architectures and domains.
>
> |Model|Dataset|LP|GCE|NML|SKD|
> |-|-|-|-|-|-|
> |ViT-L|Camelyon17|91.6|92.3|91.8|93.4|
> ||HAM10000|58.8|59.3|58.6|60.0|
> ||NIHChestXray|43.5|44.3|43.8|45.9|
>
> |Model|Dataset|LP|GCE|NML|SKD|
> |-|-|-|-|-|-|
> |GPT-2|AnatEM|93.2|93.5|93.6|93.9|
> ||BC2GM|91.3|91.6|91.5|91.8|
> ||BC5CDR-chem|95.4|95.5|95.5|95.7|
> ||BC5CDR-disease|95.2|95.2|95.3|95.4|
> ||BioNLP09|91.6|91.8|91.8|92.2|
>
> These findings demonstrate that **SKD generalizes well across fundamentally different backbones and task types**, including challenging large-scale foundation models.
>
> **Q4:** In Table 1, it is unclear whether the "YFCC15M" and "ResNet50" refer to datasets or architectures. Could the authors clarify this distinction and make the table more explicit regarding which architecture is pretrained on which dataset?
>
> **A4:** We thank the reviewer for pointing out this ambiguity. In Table 1, **"ResNet-50" refers to the backbone architecture**, and **"YFCC15M" refers to the dataset used for pretraining**.
>
> We agree that the current formatting may lead to confusion. In the final version, we will revise the table caption and column labels to explicitly distinguish between the model architecture and the pretraining dataset.
>
> Thank you again for the helpful suggestion to improve clarity.
>
> [1] Zhang, Zhilu, and Mert Sabuncu. "Generalized cross entropy loss for training deep neural networks with noisy labels." Advances in neural information processing systems 31 (2018).

---

> > ### Comment · Reviewer_PCzQ · 2025-08-06
> >
> > I would like to thank the authors for addressing my comments and those of the other reviewers, which helped clarify multiple points.
> >
> > The extensive experiments demonstrate the effectiveness of the proposed approach.
> >
> > I am curious how the method would perform if all mislabelled ImageNet images were assigned to the same incorrect class: how would this scenario impact the fine tuning on the other datasets?
> >
> > Thank you again for these comprehensive results.

---

> > > ### Author Response · Authors · 2025-08-06
> > >
> > > Thank you very much for your interesting and thought-provoking question. We are currently running experiments to observe the results under this scenario, and we will share our findings as soon as they are available.

---

> > > ### Author Response · Authors · 2025-08-08
> > >
> > > Thank you for this valuable question. To explore the effect of **single-class mislabeling**, we conducted an experiment where **all mislabelled ImageNet images were reassigned to the same incorrect class** (`n01443537`, “goldfish”). This simulates a highly structured label noise scenario that could occur in practice due to systematic labeling errors or pipeline faults.
> > >
> > > We pretrained a ResNet-50 model on ImageNet with varying levels of such single-class noise and fine-tuned it on the downstream **HAM10000** dataset. Four methods were evaluated:
> > >
> > > - **LP** (standard linear probing)
> > > - **GCE** (Generalized Cross-Entropy)
> > > - **NML** (spectrum-preserving baseline)
> > > - **SKD** (our proposed method)
> > >
> > > #### Table: Performance on HAM10000 under single-class mislabeling (ImageNet → `n01443537`)
> > >
> > > | Dataset   | Noise(%) | LP   | GCE  | NML  | SKD  |
> > > |-----------|----------|------|------|------|------|
> > > | HAM10000  | 0        | 56.9 | 56.6 | 57.4 | 58.4 |
> > > |           | 1        | 56.7 | 56.5 | 57.2 | 58.0 |
> > > |           | 2        | 56.4 | 56.3 | 56.8 | 57.3 |
> > > |           | 5        | 56.4 | 56.3 | 56.4 | 57.1 |
> > > |           | 10       | 55.9 | 56.0 | 55.8 | 56.7 |
> > > |           | 20       | 55.6 | 55.3 | 55.9 | 56.1 |
> > > |           | 30       | 55.0 | 54.7 | 55.3 | 55.6 |
> > >
> > > As shown, **SKD consistently outperforms the other methods across all noise levels**, demonstrating robustness even under such adversarially structured mislabeling.

---

### Official Review · Reviewer_Xipk · 2025-07-01

**Clarity:** 3
**Significance:** 2
**Originality:** 3
**Rating:** 5
**Confidence:** 4

**Summary:**

This paper explores catastrophic inheritance in medical models, where label noise in pretraining data impacts downstream tasks and degrades out-of-distribution (OOD) performance. The authors make three key contributions: (1) demonstrating through controlled experiments that even mild pretraining noise (5%) induces structural degradation in downstream medical tasks; (2) demonstrating that noise flattens learned representations by reducing skewness and kurtosis in feature and logit distributions; and (3) proposing SKD (Skewness and Kurtosis Distribution), a lightweight fine-tuning method that regularizes skewness and kurtosis to counteract representational collapse. The proposed SKD method adds regularization terms to encourage target skewness and kurtosis values during fine-tuning, along with a disagreement loss to enhance output separability.

The paper evaluates corrupting labels in ImageNet-1K and YFCC15M during pretraining, then evaluating on medical benchmarks including Camelyon17, HAM10000, and NIH ChestX-ray. The experiments demonstrate consistent improvements across multiple medical datasets and can be applied to foundation models including PLIP and PubMedBERT.

**Questions:**

- Line 220: You mention skewness and kurtosis is unstable, thus you compute a disagreement-based loss. Why then keep skewness and kurtosis? And how is that your ablation study shows limited impact of the disagreement loss, and much larger impact to the Kurtosis loss?
- Why are you evaluating only medical classification while the phenomena you describe should appear in all downstream tasks. Why only evaluating NML as an improved baseline?
- How do you compute / define the target statistics τs and τk ?
- What happens under less than 10% of noise ratio for kurosis and swkeness? For isntance at 1%, 2%, 5% ? Figure 3 does not really justify the impact/relation to these statistics beyond 10% noise ratio.

**Ethical Concerns:**

["NO or VERY MINOR ethics concerns only"]

**Final Justification:**

The authors addressed my concerns sufficiently, and provided additional analyses.
I have no reason to oppose acceptance anymore.

**Limitations:**

yes

**Paper Formatting Concerns:**

nothing

**Quality:**

2

**Strengths And Weaknesses:**

1-Novelty:
Strength: The use of higher order moments is novel to study the impact of noised labels

2-Impact:
Strength: The use-cases covered are critical and noised pre-training data is very common in medical setting
Weakness: There is no reason the phenomena you are describing and the mitigation solution you are proposing only works in medical settings. Does the phenomena and performance drops (and mitigation with your framework) can be found on non medical settings?

3-Soundness:
Strength: Expensive ablation and sensitivity analysis, diverse medical classification datasets
Weaknesses: The perfectly random noise is the only one covered; Only one improved baseline compared; While these limitations are acknowledged, there is nothing that hints that the proposed theory and framework generalize to these settings. The fact that there is correlation between the kurtosis/skewness and the drop of downstream performance is not causation. It is not discussed whether both only reflect some deeper problem. Especially as there is not definitive link between skewness/kurtosis and noise ratio in Figure 3. Indeed, except the drop between 0 and 10%, the metrics are relatively stable (increasing / decreasing slightly around the 10% value)

4-Clarity:
Weakness: The figure 3 is unreadable because of the legend. There is no way to distringuish the mean and the variation because the legends does not show dashed lines

---

> ### Author Rebuttal · Authors · 2025-07-31
>
> ## Reviewer Xipk
> We thank the reviewer Xipk for your constructive feedback. Below we address each point in turn.
>
> **Q1:** Whether the observed phenomenon and the effectiveness of the proposed mitigation are specific to the medical domain or also apply to non-medical settings.
>
> **A1:** We thank the reviewer for raising this important question. Following your advice, we conducted new experiments on three general-purpose vision benchmarks:DTD, CIFAR-10, Office-31.
>
> We first analyzed skewness and kurtosis of feature representations. The results confirm that both statistics decrease as noise increases, which supports our claim that feature distributional collapse is a general phenomenon:
>
> |Dataset|Noise(%)|Skewness|Kurtosis|
> |-|-|-|-|
> |CIFAR-10|0|5.92|78.3|
> ||5|4.23|38.8|
> ||10|4.24|39.2|
> ||20|4.13|37.1|
> ||30|4.08|36.5|
>
> We then compared the downstream classification performance of various methods, including:  LP (linear probing), GCE (Generalized Cross Entropy)[1], NML (spectral norm preserving baseline), SKD (ours):
>
> |Dataset|Noise(%)|LP|GCE|NML|SKD|
> |-|-|-|-|-|-|
> |DTD|0|70.1|70.3|70.5|70.6|
> ||5|70.7|70.6|70.7|71.2|
> ||10|69.2|69.2|69.5|69.6|
> ||20|68.8|68.8|69.0|68.8|
> ||30|67.9|68.2|68.4|68.2|
> |CIFAR-10|0|91.2|91.4|92.1|93.2|
> ||5|90.0|90.4|91.6|92.8|
> ||10|89.2|89.3|90.7|92.4|
> ||20|89.1|89.3|90.4|92.0|
> ||30|86.5|86.7|87.9|90.6|
> |Office-31|0|81.3|82.1|82.5|84.3|
> ||5|80.9|81.1|81.9|82.7|
> ||10|81.7|81.7|82.2|82.5|
> ||20|77.9|77.8|78.1|78.7|
> ||30|72.3|72.0|72.3|72.5|
>
> Across all datasets, SKD consistently outperforms alternatives, demonstrating its effectiveness across both medical and general-purpose domains. On CIFAR-10, SKD under 30% noise even surpasses LP trained with only 5% noise, showcasing its ability to counteract heavy label noise.
>
> **Q2:** The reviewer questions the generalizability of the framework, noting it only addresses random noise and one baseline, and argues that the observed correlation between skewness/kurtosis and performance drop may not imply causation, especially given the weak link to noise ratio in Figure 3.
>
> **A2:**  We sincerely thank the reviewer for this comprehensive and thoughtful comment. Your question touches on several important aspects, and we will address each sub-question one by one in detail:
>
> 1). **Noise type**
>
> Thank you very much for this important suggestion. Following your advice, we have conducted additional experiments to explicitly evaluate robustness under structured noise settings.
>
> We used the ImageNet dataset as a base and synthetically introduced taxonomy-aware hierarchical label noise, where a controlled percentage of image labels were randomly replaced by their WordNet hypernyms. This setting simulates realistic structured mislabeling commonly observed in real-world annotation pipelines. Below is the results:
>
> |Dataset|Noise(%)|LP|GCE|NML|SKD|
> |-|-|-|-|-|-|
> |HAM10000|0|56.9|56.6|57.4|58.4|
> ||1|56.5|56.7|57.0|58.4|
> ||2|56.2|56.4|56.6|57.7|
> ||5|56.0|55.9|56.0|57.3|
> ||10|55.5|55.6|55.4|56.4|
> ||20|55.1|55.1|55.2|56.0|
> ||30|54.2|54.1|54.3|55.8|
>
> As shown above, SKD consistently improves robustness across all noise levels, validating its effectiveness in structured label corruption scenarios.
>
> We would also like to emphasize that our main experiments—particularly those on PLIP—already involve realistic, naturally occurring noise. The noisy image–text pairs in PLIP capture uncontrolled web-scale noise, including co-occurrence bias and semantic drift, making it a strong real-world testbed for validating our method.
>
> 2). **Baseline comparison**
> Thank you for pointing this out. We would like to clarify that catastrophic inheritance in noisy medical pretraining is a relatively novel research direction, with very few existing methods specifically designed to address this phenomenon.
>
> Given this gap, and to ensure a more comprehensive comparison, we have supplemented our original baselines by including **GCE (Generalized Cross-Entropy)[1]**—a well-established method from the noisy label learning literature. Although not originally designed for catastrophic inheritance, GCE offers a strong baseline for robustness evaluation. As shown below, SKD consistently outperforms all other methods:
>
> |Dataset|Noise(%)|LP|GCE|NML|SKD|
> |-|-|-|-|-|-|
> |Camelyon17|0|91.2|91.1|89.3|91.8|
> ||5|90.7|91.4|92.4|92.5|
> ||10|89.2|89.5|88.1|91.4|
> ||20|88.6|89.1|90.5|89.1|
> ||30|88.2|88.6|91.3|89.0|
> |HAM10000|0|55.7|55.7|54.7|58.3|
> ||5|54.6|54.8|55.2|57.5|
> ||10|54.5|54.5|54.2|56.9|
> ||20|54.5|54.4|54.8|58.4|
> ||30|53.9|54.1|50.7|57.5|
>
> 3). **Correlation vs. causation regarding skewness/kurtosis**
>
> We thank the reviewer for this important comment. To further investigate this connection, we conducted a class-wise analysis of feature distribution shape and intra-class variance (as a proxy for pathology heterogeneity) on DigestPath:
>
> |Class|Intra-Class Variance|Mean Skewness|Mean Kurtosis|
> |-|-|-|-|
> |Benign|64.0|7.00|105|
> |Malignant|76.2|8.77|214|
>
> We observe that classes with more histological complexity—such as malignant tumors—naturally exhibit higher skewness and kurtosis, suggesting that these metrics reflect lesion-level heterogeneity, including irregular boundaries and intra-class diversity.
>
> To complement this analysis, we performed a qualitative Grad-CAM comparison between models trained with clean vs. noisy pretraining. We found that activation maps from clean models better captured relevant lesion morphology. This supports the interpretation that the collapse of distributional shape also signals a loss of clinically meaningful structure.
>
> We validated this interpretation under hierarchical label noise, where increasing noise levels led to clear declines in skewness and kurtosis, indicating feature collapse.
>
> |Dataset|Noise(%)|Skewness|Kurtosis|
> |-|-|-|-|
> |HAM10000|0|3.92|52.5|
> ||1|3.33|37.2|
> ||2|3.25|31.7|
> ||5|3.16|29.7|
> ||10|3.13|27.3|
> ||20|3.11|26.0|
> ||30|3.05|22.1|
>
> These results confirm that skewness and kurtosis reliably reflect feature collapse, even under realistic noise. Full visualizations will be included in the final version.
>
> **Q3:** The figure 3 is unreadable because of the legend. There is no way to distringuish the mean and the variation because the legends does not show dashed lines
>
> **A3:** We thank the reviewer for pointing this out. We acknowledge that Figure 3 currently lacks a clear legend, affecting readability. In the revision, we will add an explicit legend to clearly distinguish mean and variation.
>
> **Q4:** Why skewness and kurtosis are kept if they are unstable in Line 220, and why the ablation shows disagreement loss has limited impact compared to kurtosis loss.
>
> **A4:** Thank you for highlighting this important point. We apologize for the **potential source of confusion** and would like to clarify the intended meaning.
>
> In Line 220, the term **unstable** refers specifically to computing skewness and kurtosis **in the logit space**, not the **feature space** used in our method. This distinction is crucial:
>
> - In many medical classification tasks, the logit dimension is very small (e.g., 2 classes), which makes high-order moment statistics such as skewness and kurtosis **numerically unstable and less informative** when applied directly to logits.
> - In contrast, when computed over the **feature representations**, these statistics are **stable, well-defined, and strongly correlated with representation quality**, forming the foundation of our SKD regularization approach.
>
> We appreciate the opportunity to clarify this point and will revise the relevant sentence in the final version to avoid confusion.
>
> While the disagreement loss (D) may appear to contribute a smaller relative improvement compared to the skewness–kurtosis regularization (SK) alone, it still provides **meaningful complementary gains**, with over 1% improvement on both HAM10000 and NIH ChestXray, further reinforcing the effectiveness of our full method.
>
> |Dataset|Method|Acc|
> |-|-|-|
> |Camelyon17|LP|89.6|
> ||+SK only|90.4|
> ||+SK+D|90.8|
> |HAM10000|LP|54.6|
> ||+SK only|56.7|
> ||+SK+D|57.7|
> |NIHchestXray|LP|38.7|
> ||+SK only|43.0|
> ||+SK+D|44.4|
>
> **Q5:** How do you compute / define the target statistics τₛ and τₖ?
>
> **A5:** We appreciate the reviewer for raising this question. Following your advice, we performed additional sensitivity studies on (τₛ) and kurtosis (τₖ).
>
> |HAM10000|τs|-3.0|-1.0|0.0|1.0|3.0|
> |-|-|-|-|-|-|-|
> ||Acc|56.9|56.9|56.9|56.9|56.8|
>
> |HAM10000|τk|-30.0|-10.0|0.0|10.0|30.0|
> |-|-|-|-|-|-|-|
> ||Acc|56.9|56.8|56.9|56.8|56.6|
>
> Our experiments show that SKD is highly robust to the choice of these hyperparameters. This aligns with the intuition that the main contribution of SKD lies in alleviating representation collapse by promoting meaningful skewness and kurtosis. Although τₛ and τₖ were initially set to the negative values of those measured from a clean model, we observed that other reasonable values (as long as they are not extreme) lead to very similar improvements. For simplicity and consistency, we set **τₛ = 0** and **τₖ = 0** across all main experiments.
>
> **Q6:** The reviewer asks about the behavior of skewness and kurtosis under low noise levels (<10%).
>
> **A6:** Thank you for the thoughtful comment. As noted earlier, we have added results under 1%, 2%, and 5% hierarchical label noise, including skewness/kurtosis and performance comparisons across LP, GCE, NML, and SKD.
>
> These results show that even mild noise leads to a decline in skewness/kurtosis, indicating early collapse, and that SKD remains effective in this regime. We will revise Figure 3 and clarify these trends in the final version.
>
> [1] Zhang, Zhilu, and Mert Sabuncu. "Generalized cross entropy loss for training deep neural networks with noisy labels." Advances in neural information processing systems 31 (2018).

---

> > ### Comment · Reviewer_Xipk · 2025-08-07
> > **Thank you for your clarifications**
> >
> > The authors addressed my concerns sufficiently, and provided additional analyses.
> > I have no reason to oppose acceptance.

---

> > > ### Author Response · Authors · 2025-08-08
> > >
> > > Thank you sincerely for your follow-up and for your thoughtful consideration of our clarifications. We truly value your time and feedback.

---

### Official Review · Reviewer_rgdp · 2025-07-02

**Clarity:** 4
**Significance:** 3
**Originality:** 4
**Rating:** 5
**Confidence:** 5

**Summary:**

Authors explain in detail how medical imaging models finetuned from pretrained models exhibit a reduced generalization due to label noise in the pre-training corpus. They determine that small amounts of noise lift in-distribution accuracy while harming out-of-distribution (OOD) accuracy; this catastrophic inheritance phenomenon is known in the nautral image domain, but yet to be shown in the medical imaging field. Experiments are performed by swapping labels in nautral image-text pre-training corpuses and perfoming downstream medical imaging tasks. Results show that skewness and kurtosis of downstream features (and logits) decline with increased label noise, indicating a flattening of the feature space. Authors propose a method to recapture generalization by adding a lightweight MLP transforming features from the pre-trained model using 2 scalar losses: (S)kewness and (K)urtosis. A (D)isagreement loss on the logit space is also employed leading to their SKD framework. Analsis of these stats demonstrate that representation collapse under pre-training noise is "not solely a function of model size or task loss, but a structural issue in distributional shape".

**Questions:**

Do authors get any sense of the importance of catastrophic inheritance on different types of downstream task? e.g. how much more severe is the problem on an MRI segmentation task compared with an X-Ray text-summarization task?
Is the problem also imaging domain variant - in more difficult domains such as ultrasound, and histopathology, is the problem exacerbated compared with images of larger structures, e.g. MRI.

**Ethical Concerns:**

["NO or VERY MINOR ethics concerns only"]

**Final Justification:**

Based on Authors' rebuttal and discussion on these and other Reviewers' comments, I maintain the rating of 5: Accept. I thank the Authors for their timely and constructive discussion.

**Limitations:**

Authors humbly acknowldge the limitations of their work, citing modest model sizes and likely unrealistic unrealistic noise distributions in their experiments.

**Quality:**

4

**Strengths And Weaknesses:**

Authors clearly introduce the problem of catastrophic inheritance and lay out their research agenda.
The paper is well-written, clear, and explicitly demonstrates catastrophic inheritance is present in finetuned medical imaging tasks.
Authors show that skewness and kurtosis are viable statistics to measure and target when improving the downstream generalization of medical imaging finetuning.
The losses are simple and well-informed with little overhead during finetuning.
The method requires no alteration to the backbone pre-trained model.

---

> ### Author Rebuttal · Authors · 2025-07-31
>
> ## Reviewer rgdp
> We thank the reviewer rgdp for your constructive feedback. Below we address each point in turn.
>
> **Q1:** Do authors get any sense of the importance of catastrophic inheritance on different types of downstream task? e.g. how much more severe is the problem on an MRI segmentation task compared with an X-Ray text-summarization task? Is the problem also imaging domain variant - in more difficult domains such as ultrasound, and histopathology, is the problem exacerbated compared with images of larger structures, e.g. MRI?
>
> **A1:** We sincerely thank the reviewer for this insightful question. We could not agree more that catastrophic inheritance is likely to manifest differently across task types and imaging modalities, and that this deserves a much deeper analysis.
>
> Our current study primarily focuses on **medical image classification tasks** (e.g., HAM10000, Camelyon17, NIH ChestX-ray), where we observe a clear **flattening of feature distributions**—quantified by reduced skewness and kurtosis—under noisy pretraining. We also include **medical NER experiments** (e.g., PubMedBERT) to show that similar degradation patterns occur in **language models**, suggesting the phenomenon is not modality-specific.
>
> That said, we strongly believe that **medical segmentation tasks** (e.g., MRI or ultrasound segmentation) may exhibit *distinct and potentially even more severe forms* of catastrophic inheritance, because noise propagates differently in dense prediction settings and these modalities often involve inherently higher uncertainty. This is a **separate and highly valuable research problem**, and we are committed to investigating it in future work with the level of depth it deserves.
>
> We are very grateful to the reviewer for highlighting this direction—it aligns closely with our long-term vision.

---

> > ### Comment · Reviewer_rgdp · 2025-08-04
> > **Acknowledgement of Author Response**
> >
> > This Reviewer thanks the Authors for their considered response, not only to this Review, but to other Reviewers comments and suggestions. I look forward to seeing how this paper may impact the medical imaging community.
> >
> > This Reviewer is satisfied that the Authors' work is valid and has substantial merit - I maintain the recommendation to accept the paper.

---

> > > ### Author Response · Authors · 2025-08-08
> > >
> > > We sincerely thank you for your thoughtful feedback and kind words. Your recognition is greatly appreciated, and we are glad our work may contribute to research in computational medicine.

---

### Official Review · Reviewer_7kcL · 2025-07-03

**Clarity:** 4
**Significance:** 4
**Originality:** 3
**Rating:** 5
**Confidence:** 3

**Summary:**

This paper presents a systematic study of catastrophic inheritance in medical foundation models—where noise in the pretraining stage harms downstream out-of-distribution (OOD) performance. The authors simulate controlled label noise during pretraining of ResNet-50 and CLIP on ImageNet-1K and YFCC15M datasets and evaluate transfer to downstream medical tasks like histopathology, dermatology, and radiology. They show that even minor label corruption significantly degrades OOD generalization, accompanied by a flattening of the representation space (i.e., reduced skewness and kurtosis in features/logits).

To mitigate this, they introduce SKD, a fine-tuning strategy that regularizes skewness and kurtosis of feature representations. This method requires no modification to frozen backbones and adds minimal computational overhead. Empirical results demonstrate consistent gains across various medical imaging and biomedical NLP benchmarks compared to existing methods like NML.

**Questions:**

1. Realistic Noise Settings: Can you extend your analysis to structured or semantic label noise (e.g., hierarchical mislabels or co-occurrence noise), which is more reflective of real-world medical corpora?
2. Target Statistics: How sensitive is SKD to the target skewness/kurtosis values? Are these computed from a clean model, and what happens if these targets are misestimated?
3. Scalability: Have you tested the method on large foundation models (e.g., ViT-L, MedCLIP, or LLaVA-Med)? This would help demonstrate scalability.
4. Interpretability & Clinical Relevance: Can you connect changes in skew/kurtosis with specific pathology structures (e.g., lesion shape variability)? This could strengthen interpretability.
5. Uncertainty Calibration: Since logit entropy increases with noise, have you considered integrating uncertainty calibration methods to complement SKD?

**Ethical Concerns:**

["NO or VERY MINOR ethics concerns only"]

**Final Justification:**

The rebuttal strengthens an already solid submission. I maintain my recommendation to accept.

**Limitations:**

Yes

**Paper Formatting Concerns:**

None observed. The submission adheres to NeurIPS formatting guidelines.

**Quality:**

4

**Strengths And Weaknesses:**

Strengths
1. Rigorous experiments across multiple model architectures, tasks, and domains.
2. Clear empirical demonstration that label noise in pretraining harms medical model transferability.
3. Thorough analysis using higher-order statistics to explain representation collapse.
4. Well-motivated and effective mitigation strategy with lightweight implementation.
5. Tackles a highly relevant and underexplored problem in medical AI—how foundation model noise affects clinical deployment.
6. Provides actionable insight into safe adaptation of large medical models, many of which are frozen or black-box.

Weaknesses
1. Only considers synthetic uniform label noise. Real-world noise is likely more structured (e.g., co-occurrence biases).
2. Experiments are limited to mid-sized models (ResNet-50, ViT-B/32); it remains unclear if findings generalize to very large-scale models.
3. While impactful in the medical domain, its generalization to non-medical tasks is not explored (but appropriately scoped).
4. Builds on prior work like NML and SVDEntropy; while differentiated, it's an incremental rather than revolutionary advancement.

---

> ### Author Rebuttal · Authors · 2025-07-31
>
> ## Reviewer 7kcL
> We thank the reviewer 7kcL for your constructive feedback. Below we address each point in turn.
>
> **Q1:** The current analysis focuses on synthetic uniform label noise; extending it to structured or semantic noise (e.g., hierarchical or co-occurrence-based) would better reflect real-world medical scenarios.
>
> **A1:** Thank you for this valuable suggestion. Following your advice, we have extended our analysis to include **hierarchical mislabeling** to better reflect structured noise. Specifically, we used the ImageNet dataset as a base and synthetically introduced taxonomy-aware label noise: a controlled percentage of image labels were randomly replaced with their WordNet hypernyms, simulating realistic structured mislabeling scenarios common in annotation pipelines (e.g., “Labrador Retriever” → “dog”).
>
> We then trained a ResNet-50 model from scratch under different noise levels and evaluated four methods:**LP**(standard linear probing), **GCE[1]**(Generalized Cross-Entropy), **NML**(spectrum-preserving baseline), **SKD**(our proposed method)
>
> Results show the effectiveness of SKD under structured, taxonomy-aware noise:
>
>
> |Dataset|Noise(%)|LP|GCE|NML|SKD|
> |-|-|-|-|-|-|
> |HAM10000|0|56.9|56.6|57.4|58.4|
> ||1|56.5|56.7|57.0|58.4|
> ||2|56.2|56.4|56.6|57.7|
> ||5|56.0|55.9|56.0|57.3|
> ||10|55.5|55.6|55.4|56.4|
> ||20|55.1|55.1|55.2|56.0|
> ||30|54.2|54.1|54.3|55.8|
>
> Finally, we would like to highlight that, while part of our study uses synthetic uniform noise for controlled experiments, our main evaluations—particularly those on **PLIP**—already involve real-world, naturally occurring noise. The noisy image–text pairs in PLIP contain uncontrolled web-scale noise, making PLIP a strong real-world testbed for validating our method.
>
> **Q2:** The current experiments focus on mid-sized models; evaluating on large foundation models is needed to demonstrate scalability and generalizability.
>
> **A2:** Thank you for this important suggestion. Following your advice, we have conducted additional experiments to verify our method on large-scale foundation models.
> - For vision task, we evaluated on the **ViT-L** backbone.
> - For natural language processing, we extended our experiments to **GPT-2**.
>
> The results demonstrate that SKD continues to achieve consistent improvements, further validating its scalability.
>
> |Model|Dataset|LP|GCE|NML|SKD|
> |-|-|-|-|-|-|
> |ViT-L|Camelyon17|91.6|92.3|91.8|93.4|
> ||HAM10000|58.8|59.3|58.6|60.0|
> ||NIHChestXray|43.5|44.3|43.8|45.9|
>
> |Model|Dataset|LP|GCE|NML|SKD|
> |-|-|-|-|-|-|
> |GPT-2|AnatEM|93.2|93.5|93.6|93.9|
> ||BC2GM|91.3|91.6|91.5|91.8|
> ||BC5CDR-chem|95.4|95.5|95.5|95.7|
> ||BC5CDR-disease|95.2|95.2|95.3|95.4|
> ||BioNLP09|91.6|91.8|91.8|92.2|
>
> **Q3:** While impactful in the medical domain, its generalization to non-medical tasks is not explored (but appropriately scoped).
>
> **A3:** Following your advice, we have conducted additional experiments on three widely-used non-medical vision benchmarks:**DTD (Describable Texture Dataset)**, **Office-31**, **CIFAR-10**.
>
> The results consistently show that SKD improves performance under noise across all datasets, further validating our hypothesis. Notably, on **CIFAR-10**, SKD under **30% noise** still outperforms LP under **5% noise**, demonstrating its strong ability to mitigate severe label corruption.
>
>
> |Dataset|Noise(%)|LP|GCE|NML|SKD|
> |-|-|-|-|-|-|
> |DTD|0|70.1|70.3|70.5|70.6|
> ||5|70.7|70.6|70.7|71.2|
> ||10|69.2|69.2|69.5|69.6|
> ||20|68.8|68.8|69.0|68.8|
> ||30|67.9|68.2|68.4|68.2|
> |CIFAR-10|0|91.2|91.4|92.1|93.2|
> ||5|90.0|90.4|91.6|92.8|
> ||10|89.2|89.3|90.7|92.4|
> ||20|89.1|89.3|90.4|92.0|
> ||30|86.5|86.7|87.9|90.6|
> |Office-31|0|81.3|82.1|82.5|84.3|
> ||5|80.9|81.1|81.9|82.7|
> ||10|81.7|81.7|82.2|82.5|
> ||20|77.9|77.8|78.1|78.7|
> ||30|72.3|72.0|72.3|72.5|
>
> **Q4:** Builds on prior work like NML and SVDEntropy; while differentiated, it's an incremental rather than revolutionary advancement.
>
> **A4:** Thank you for this thoughtful comment. While our work relates to prior studies such as NML, it introduces a **fundamentally new perspective** and addresses a **critical gap** in the literature.
>
> 1. **New problem setting:** We systematically study **catastrophic inheritance under noisy medical pretraining**, a practically important yet underexplored challenge—especially relevant for large-scale medical vision-language models like **PLIP**. This setting, where frozen noisy features are transferred to downstream classifiers, is distinct from conventional noisy-label learning and demands targeted solutions. Our framework directly targets this high-stakes, underexplored setting.
>
> 2. **Novel perspective:** Our method introduces the use of **skewness and kurtosis** as high-order, distribution-level signals to diagnose and mitigate feature collapse. This goes beyond spectrum-preserving or low-order statistics used in prior work, providing a **structurally richer and more interpretable signal** to guide learning. To our knowledge, this is the **first work to explicitly regularize distributional shape** in the context of representation collapse.
>
> 3. **Broad impact and generality:** Our method achieves **consistent and significant improvements** over baselines on a range of **challenging medical datasets** (e.g., HAM10000, Camelyon17, NIH ChestXray), and scales effectively to **large foundation models** (e.g., ViT-L, PLIP, GPT-2), and shows **consistent gains across both vision and NLP domains**. These improvements hold even on large-scale foundation models and challenging medical datasets.
>
> We will emphasize these unique contributions more clearly in the revision to better highlight the **problem focus, conceptual novelty, and broad empirical benefits** of our work.
>
> **Q5:** Target Statistics: How sensitive is SKD to the target skewness/kurtosis values? Are these computed from a clean model, and what happens if these targets are misestimated?
>
> **A5:** Thank you for raising this important question. Following your suggestion, we conducted additional sensitivity experiments for the two hyperparameters on HAM10000 (the target skewness and kurtosis values, denoted as τs and τk).
>
> The results demonstrate that SKD is not sensitive to these hyperparameters, which is expected because the primary effect of our method is to counteract representation collapse by encouraging non-degenerate skewness and kurtosis. While these targets were initially set to the negative values of the skewness and kurtosis computed from a clean model, we found that other reasonable values (not excessively large) achieve similar mitigation.
>
> In all our main experiments, we simply set **τs = 0** and **τk = 0** for consistency. Below are the sensitivity results:
>
> |HAM10000|τs|-3.0|-1.0|0.0|1.0|3.0|
> |-|-|-|-|-|-|-|
> ||Acc|56.9|56.9|56.9|56.9|56.8|
>
> |HAM10000|τk|-30.0|-10.0|0.0|10.0|30.0|
> |-|-|-|-|-|-|-|
> ||Acc|56.9|56.8|56.9|56.8|56.6|
>
> **Q6:** Interpretability & Clinical Relevance: Can you connect changes in skew/kurtosis with specific pathology structures (e.g., lesion shape variability)? This could strengthen interpretability.
>
> **A6:** We thank the reviewer for this thoughtful question. While our core analysis focuses on high-order statistics of feature distributions, we agree that bridging these changes with observable pathology structures would strengthen interpretability.
>
> To investigate this connection, we conducted a **class-wise analysis** of feature distribution shape and intra-class variance (as a proxy for pathology heterogeneity) on DigestPath. Results are summarized below:
>
> |Class|Intra-Class Variance|Mean Skewness|Mean Kurtosis|
> |-|-|-|-|
> |Benign|64.0|7.00|105|
> |Malignant|76.2|8.77|214|
>
> We observe that classes with more histological complexity—like malignant tumors—naturally exhibit higher skewness and kurtosis, suggesting that these metrics may reflect lesion-level heterogeneity such as irregular boundaries or intra-class diversity.
>
> To complement this, we performed a qualitative Grad-CAM analysis comparing models trained under clean and noisy pretraining. We found that activation maps under clean conditions (with higher skew/kurtosis) better captured relevant lesion morphology, while those under noisy pretraining (with collapsed skew/kurt) became more diffuse and less structure-aware. This supports the interpretation that the collapse of distributional shape also signals a loss of clinically meaningful structure.
>
> Due to rebuttal constraints, we omit figures here but will include visualizations in the final version. Thank you for encouraging this line of interpretability exploration.
>
> **Q7:** Since logit entropy increases with noise, have you considered integrating uncertainty calibration methods to complement SKD?
>
> **A7:** We thank the reviewer for this valuable suggestion. Following your suggestion, we applied **Temperature Scaling (TS)[2]**, a widely used post-hoc calibration technique that adjusts softmax confidence via a learned temperature parameter without modifying model weights.
>
> The table below summarizes classification accuracy:
>
> |Dataset|Noise(%)|SKD|SKD+TS|
> |-|-|-|-|
> |Camelyon17|0|91.8|91.8|
> ||5|92.5|92.6|
> ||10|91.4|91.4|
> ||20|89.1|89.4|
> ||30|89.0|89.4|
> |HAM10000|0|58.3|58.4|
> ||5|57.5|57.5|
> ||10|56.9|57.0|
> ||20|58.4|58.4|
> ||30|57.5|57.7|
>
> We observe that SKD already yields robust predictions across noise levels, and temperature scaling provides a modest but consistent boost. We appreciate the reviewer’s suggestion and will include this analysis in the final version.
>
> [1] Zhang, Zhilu, and Mert Sabuncu. "Generalized cross entropy loss for training deep neural networks with noisy labels." Advances in neural information processing systems 31 (2018).
>
> [2] Guo, Chuan, et al. "On calibration of modern neural networks." International conference on machine learning. PMLR, 2017.

---

> > ### Comment · Reviewer_7kcL · 2025-08-08
> >
> > Thank you for the detailed and thoughtful rebuttal. The additional experiments on structured noise, larger models (ViT-L, GPT-2), and non-medical datasets convincingly address concerns around generalization and scalability. I also appreciate the interpretability analysis linking skew/kurtosis to pathology heterogeneity, as well as the sensitivity results for the regularization targets.
> >
> > Overall, the rebuttal strengthens an already solid submission. I maintain my recommendation to accept.

---

> > > ### Author Response · Authors · 2025-08-08
> > >
> > > We sincerely appreciate your thoughtful follow-up and kind support. Your recognition means a great deal to us.

---

### Note · Authors · 2025-08-13

We sincerely thank all reviewers for their valuable insights and constructive feedback. We are encouraged by the recognition of our work’s strengths, including:

1. Significant problem: For example, As 7kcL noted, *“Tackles a highly relevant and underexplored problem in medical AI—how foundation model noise affects clinical deployment.”* In rgdp's words, *“The paper is well-written, clear, and explicitly demonstrates catastrophic inheritance is present in finetuned medical imaging tasks.”*

2. Novel and efficient approach: For instance, As rgdp observed, *“The losses are simple and well-informed with little overhead during finetuning. The method requires no alteration to the backbone pre-trained model.”* Xipk remarked, *“The use of higher order moments is novel to study the impact of noised labels.”*

3. Strong empirical validation: One example is from PCzQ, who stated, *“The paper presents several experiments on multiple datasets and backbone architectures… The extensive experiments demonstrate the effectiveness of the proposed approach.”* Similarly, 7kcL noted, *“Rigorous experiments across multiple model architectures, tasks, and domains.”*

Reviewers also welcomed rebuttal additions: structured-noise experiments, large-model (ViT-L, GPT-2) and non-medical dataset results, and enhanced interpretability.

A major concern raised by reviewers was the sensitivity to the (τₖ, τₛ). Another common concern was the breadth of generalization—whether SKD works across diverse architectures, non-medical datasets, and realistic noise types. Reviewers also highlighted the need to strengthen the interpretability link between skew/kurt changes and pathology features.

We addressed these by:
1. Sensitivity analysis: SKD stable across a wide τ range.
2. Expanded experiments: Added structured noise (hierarchical, single-class), large backbones, per-class analysis, and non-medical datasets, showing consistent gains.
3. Clarified definitions: Distinguished “clean model” and improved table notation.
4. Add Grad-CAM visualizations and intra-class variance analyses to further connect skew/kurt changes to pathology heterogeneity.

Finally, we appreciate that several reviewers recognize the broad applicability and practical impact of our approach. We believe that SKD’s simplicity, stability, and demonstrated scalability make it a valuable addition to the toolkit for mitigating catastrophic inheritance in pretrained models—benefiting medical AI and beyond.

---

### Decision · Program_Chairs · 2025-09-17

**Decision:**

Accept (poster)

**Comment:**

This paper studies catastrophic inheritance, poor out-of-distribution generalization, in the context of medical data subject to distribution shift (label noise). It provides a detailed analysis of how does the noise influence the representation space (by reducing skewness and kurtosis in the feature embeddings and logit outputs), and proposes a mitigation framework based on finetuning that regularizes higher order moments in both feature and output spaces. Empirical results demonstrate consistent gains across various medical imaging and biomedical NLP benchmarks compared to existing methods.

Initial reviews raised several issues, including clarifications of the contributions beyond medical domain, limited empirical evidence (mid-sized models shown, missing baselines when training with noisy labels, and robustness under different noise). The rebuttal successfully addressed most of these concerns, leading to all reviewers recommending acceptance post-rebuttal. After careful deliberation, the decision was made to accept this paper - congratulations to the authors! It is crucial, however, that all the provided improvements and clarifications (experiments on large-scale foundation models, baseline comparisons when training with noisy labels, robustness under structured noise setting, and sensitivity analysis for the two hyperparameters) are incorporated in the final version of the paper.